# REDHAT 🎅: REDUCING HALLUCINATION IN ESSAY CRITIQUES FROM LARGE LANGUAGE MODELS

## ABSTRACT

Essay critiques refer to the textual assessment of an essay, serving as the basis for the grading of the essay, and are also crucial for the improvements of the essay. Essay critique generation has received increasing attention after the blooming of large language models (LLMs), which show promising potential in writing and critiquing essays. However, current LLMs suffer from hallucinations when generating essay critiques (e.g., baseless criticism), which are still under-explored in the community. To facilitate research in reliable essay critique generation, we first define this task with a unified input-output format as well as clear evaluation criteria. To minimize hallucinations in critique generation, we introduce RedHat, a novel approach that embeds the key information from an essay directly into the generation process through document-level question-answering, ensuring critiques stay firmly anchored to the evaluated content. We collected a large-scale, high-quality essay critique dataset called EssayC, annotated by human experts over multiple LLM-generated critiques, from an undergraduate essay writing course. We experimented RedHat backboned by commercial and open-sourced LLMs. Results showed that critiques generated by RedHat are preferred by auto-judger and human experts over baseline in around 20% of cases on EssayC in ambiguity and informativeness, with a decrement around 10% on hallucinations under our evaluation criteria.

## 1 INTRODUCTION

Essay critiques are pivotal for grading writings (Triawan et al., 2023; Suresh et al., 2023; Wang et al., 2018), providing constructive feedback (Abbas & Herdi, 2018) and improving writing skills (Noroozi et al., 2023). With the advancement of large language models (LLMs) (Ouyang et al., 2022; Rafailov et al., 2024; Ethayarajh et al., 2024), LLM-as-a-judge (Zheng et al., 2024a) based critique models have shown promising results in providing explainable and informative critiques in instruction following tasks (Ke et al., 2023; OpenAI, 2024a). Although applying LLMs in essay assessment seems promising, our study found that LLMs are plagued by hallucinations when generating essay critiques and therefore not suitable for direct application.

Hallucination in LLMs refers to the phenomenon that the generated content is not grounded on factual or correct information (Rawte et al., 2023). Figure 1 presents hallucinations from GPT-4o (OpenAI, 2024b) generated essay critiques. It exhibits two typical types of hallucination in this task: (1) providing advice that is not appropriate nor does not match the essay content, and (2) proposing fallacies that do not exist in the assessed essay. These hallucinated critiques significantly hinder the usability of LLM in essay critique generation.

Existing research focuses on instructing LLMs to automated essay scoring (AES) (Kundu & Barbosa, 2024), yet improving critique quality is still under-explored. Lack of consensus on how to evaluate an essay in detail leads to such negligence in critique improvement. First, the essay is a form of open-ended generation (Brahman et al., 2022), ranging from narrative to argumentative, each with distinct purposes. Detailed requirements differ between writing an analysis part and a conclusion part. This complicates the detailing of assessment criteria in the evaluation prompt, leading to the fact that type I hallucination in Figure 1 often happens. Unfortunately, human expert evaluation is extremely costly and inefficient (10 seconds for LLMs versus half an hour for human) for assess-

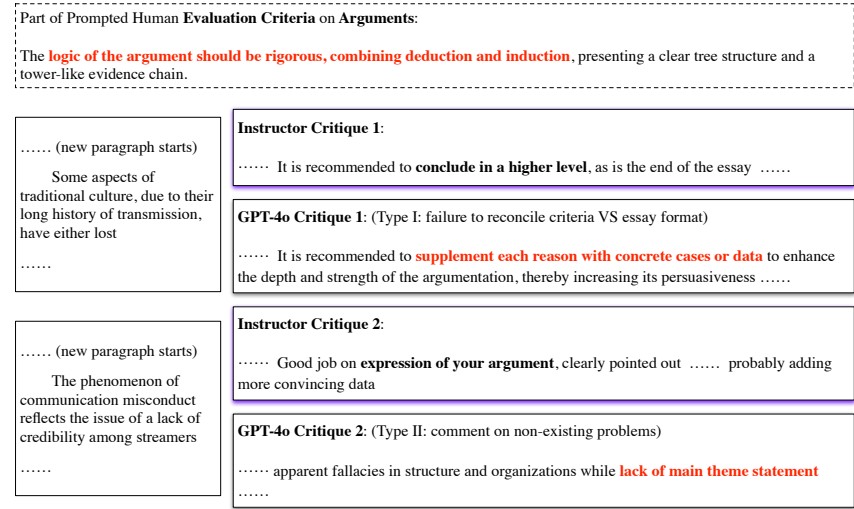

Figure 1: **Essay critique hallucination explained**. Here we listed two typical hallucinations caused by LLM's overly following evaluation criteria in the prompt of the whole essay when using GPT-4o-mini to generate essay critique. In the case of **Type I** hallucination, we find that GPT-4o-mini is overly criticizing a conclusion part using descriptions from the criteria. In the case of **Type II** hallucination, GPT-4o-mini does not capture the author's argument expressed in complex format and presented in the beginning. GPT-4o-mini is fed with the whole essay and criteria. The prompt for generating critique is listed in Appendix I.

ment both for essays and critiques, causing a lack of research resources, especially for the detection of Type II hallucination in Figure 1. These factors hinder the understanding and de-hallucination of LLM-generated essay critiques.

In response to the lack of resource challenges, we start with an undergraduate writing training course. We regard such scenarios as a generalizable scene of essay critiquing, with abundantly available expert annotations. In our preliminary study of the application of LLMs in such a course, we found that the LLM hallucination behavior in essay critiques is closely related to two factors: (1) misinterpretation of the essay, and (2) over-exaggeration of a certain standard in the criteria. These factors plague the LLM with hallucinations in following the critiquing prompt.

Based on the findings above, we propose RedHat (Reduce HallucinaTion), to reduce the loss of credibility in LLM-generated essay critiques caused by hallucination. We noticed a widely adopted assessment technique for humans (Marton & Säaljö, 1976; Chung et al., 2023), that embodies the abstract concept of understanding an essay into the concrete practice of answering questions based on it. We consulted essay experts to identify crucial questions that often need to be clarified for a thorough understanding of an essay. The answers to these questions were beneficial to facilitate the LLM's comprehension of the essay's structure and the author's arguments. This approach ensures that the model recognizes the function of each paragraph within the overall context of the essay, reducing the risk of generating hallucinations caused by overly following evaluation instructions. These question-answer pairs were embedded into the evaluation prompt, providing additional grounding information for the essay during the generation process

We compare accessible alignment techniques including post-pretraining on long contexts and supervised finetuning with RedHat. We suggest the generalizability of RedHat across different base-LLMs, languages and writing genres. We show that alignment would cause more hallucinations on synthetic training data constructed out of human experts' critiques. This indicates the source of such new hallucinations. In our evaluation setting, RedHat augmented LLM is consistently preferred by human annotators compared to baselines. We utilize the optimized critiques as guidance for essay improvement. In our machine-aided refinement setting, the polished content is generally preferred by human annotators. These showed the potential of our method in relieving hallucination in critiques, thus providing essays with informative and practicable help.

| Works | Granularity | Target | Content Len | Generation Format | Generation Len | Open sourced |
|---|---|---|---|---|---|---|
| Ours | Paragraph | Argumentative Writing | 5K | Critique | ∼100 | ✔ |
| (Tyser et al., 2024) | Whole | CS Conference Paper | >10K | Review | Unlimited | ✘ |
| (Liu & Shah, 2023) | Whole | CS Conference Paper | >10K | Review | Unlimited | ✔ |
| (Tang et al., 2024) | Whole | ASAP-AES[1] | 150-550 | Score | Integer | ✔ |
| (Noroozi et al., 2023) | Sentence | Argumentative Writing | <800 | Feedback | 30-50 | ✘ |

Table 1: **A Brief Comparison with Previous Work.** We conclude between the scope that AI feedback covers(**Granularity**), assessment content (**Target**), content length (**Writing Len**), AI feedback format, length and whether the works' dataset, method and evaluation results are publicly available (**Open Sourced**). Our work integrates a fine-grained perspective towards this field.

## 2 FINE-GRAINED ESSAY CRITIQUE GENERATION

### 2.1 CREATION OF EssayC

Our task is to leverage Large Language Models (LLMs) to automate the generation of critiques for the drafts of undergraduates' argumentative essays. The goal is to provide students with meaningful feedback that aligns with the detailed feedback provided by the instructors, so as to help the students improve their writing before final submission (**critiquing**). Most previous works studies whether LLM feedback for essay **scoring** (Tang et al., 2024), distinguish the quality of paired abstract (Liu & Shah, 2023) or **trait scoring** in conference review (Tyser et al., 2024). Our evaluation criteria including topic, literature, arguments and structure, language, and norms can be inspected in Appendix C. As summarized in Table 1, this work distinguishes itself from previous work mainly from types and granularity from the LLMs, the assessed target and more importantly, the shift from scoring to critiquing.

In order to provide a unified, even and reproducible testbench for such task, we curated EssayC addressing such task and concerns. EssayC randomly collects undergraduate essays whose topics cover *Environment Science, Biological Science, Software Engineering, Game Industry, Earth, Social Science, Journalism and Communications, Economics, Humanities, Literature Comments*, and so on. Most science, engineering and humanity and social science are covered. Human comments may be incomplete in grammar and organization. We used GLM4-130B to refine and complete their grammar and structure based on the human comments. As a result, 36 essays are randomly picked out of the above process under each field topics. The leftovers are beneficial as training data for supervised fine-tuning.

To enhance quality, we asked the annotators to read through the teacher's critiques in the paragraph and filter out unqualified ones, such as those with only punctuation marks or subjective comments expressing unrelated feelings. Then we devised a raw critique-quality classifier on GLM-4-9B to auto-filter the leftovers in the train data section. Critique numbers drop from 675 to 395 in the test set, and from 51238 to 31694 in the training set after filtration. The statistics can be found in Table 2 column 2.

### 2.2 ESSAY CRITIQUE GENERATION TASK DESCRIPTION

Paragraph-level feedback is an effective granularity for improving written content since it can effectively help authors localize the problem while maintaining most contextual information. In contrast, sentence-level feedback can sometimes be less meaningful because not every sentence in a text

|  | EssayC | English |
|---|---|---|
| Essays | 36 | 10 |
| Avg Len | 5204.7 | 42087.3 |
| Critiqued Paras | 395 | 100 |
| Avg Para Len | 278.2 | 1278.4 |
| Avg Tea Cri Len | 76.78 | / |
| Pointwise Annotations | 5530 | 200 |
| Annotation Dims | 4 | 4 |
| Pairwise Annotations | 1580 | 100 |
| Avg Cri len | 98.53 | 89.65 |

Table 2: **Statistics about** EssayC. **Avg** is short for average. **Para** is short for paragraphs. **Tea** is short for teachers. **Cri** is short for critique. **English** stands for the English subset of conference papers used in experiment.

requires revision, nor are writing problems typically confined to a single sentence. We focus on essay assessment for pedagogical purposes, mainly helping novice writers improve their writing with detailed and informative critiques, and we formulate the task as follows.

**Task Formalization**: Given an essay $\mathcal{E}$, a set of instructor evaluation criteria $\Gamma$, and the paragraph $\mathcal{P}$ to be critiqued, a model $f$ (e.g., an LLM) is required to generate a critique $c$ for that paragraph:

$$c = f(\mathcal{E}, \Gamma, \mathcal{P}) \tag{1}$$

**Objective**: The goal of this task is to generate critiques that meet three essential criteria. First, the critique should be free from **hallucination**, and accurately interpret the author's viewpoints and the factual evidence in the text without introducing inaccuracies. Second, it must be **detailed**, demonstrating a thorough understanding of the paragraph under evaluation, rather than providing vague or superficial feedback. Finally, the critique should be **informative**, offering meaningful insights that assist authors in improving their writing. To maintain clarity and readability, we stipulate that comments must be limited to a maximum of 100 words in our study. Formally, the generation of critique $c$ should maximize the **informativeness** $\mathcal{U}(c)$ and **Ambiguity** $A(c)$ while minimizing **hallucination** $H(c)$, subject to the length constraint:

$$\max_c \quad \mathcal{U}(c) - A(c) - H(c), \quad \text{subject to} \quad \text{Len}(c) \le 100 \tag{2}$$

This constrained problem reflects the trade-off between reducing hallucinations and increasing detail, with the ultimate goal of optimizing the informativeness of the feedback provided to the writer.

## 3 HALLUCINATION IN ESSAY ASSESSMENT

We conducted an empirical study using students' feedback on LLM-generated critiques. Students give textual judgments over randomly presented critiques to their essays generated by LLMs including ChatGLM3-6b, GLM-4 Plus API (Du et al., 2021; GLM, 2024), and ChatGLM3 finetuned on the instructors' comments. We found that most prominent issue is hallucination in critiques, as reported to be "the generation of plausible looking yet factually incorrect statements" from (Bang et al., 2023).

As (Maynez et al., 2020) defined *Extrinsic Hallucination* as "ignoring the source material altogether" and *Intrinsic Hallucination* as "misrepresenting information from the document" in summarization task, we found the hallucination in generated essay critiques can be divided mainly into two types as follows:

- **Type I**: Criticizing writing fallacies that do not exist in the essay. As the cases in Table 3 show, LLM emphasize some baseless errors. This type shares commons with the above *Extrinsic Hallucination*.

- **Type II**: Overemphasizing details and reversal of priorities in argumentation structures. The primary concern lies in the tendency to recommend inclusion of excessive details, which consequently undermines the clarity and conciseness of the argument. This diverts from the actual intent proposed in the criteria and the essay. This type is partly related to *Intrinsic Hallucination*.

Under the two main types of hallucinations, we discuss the specific manifestation of them. As listed in Table 3, ignoring the context information, or misunderstanding authors' perspective originate from **Type I** hallucination. Overreaction and Over-elaboration originate from **Type II** hallucination. These consist of the major aspects for human judgment of critique quality in experiments, such as Table 4 and Appendix H.

We also observed that the occurrence of hallucination varies depends on the position of the critiqued content within the essay. The conclusion part of the essay exhibits the least amount of hallucination, whereas the body sections exhibit the highest incidence. Figure 3 illustrates the human scoring of critique quality, primarily based on the extent of hallucination. Hallucinations are most pronounced in the essay sections ranging from positions between 0% to 30% and around 80% with respect to the total essay length, indicating that LLMs struggle particularly in these areas.

| Hallucination | Description | Example Cases | Type |
|---|---|---|---|
| Ignoring context info | Overlooked the contextual information, failing to notice the perspectives and evidence the author has already provided in the surrounding text | (**critiquing conclusion part**) This section provides background information on "carnivalization"; however, it is somewhat lacking in argumentation and support for the viewpoint. | Type I |
| Overcorrection at the word or sentence level | Incorrect correction of words or phrases, or overcorrection | In addition, the argument lacks detailed support , and terms such as "universality" and "social attributes" are not thoroughly explained. (**no need for explanation**) | Type II |
| Misunderstand the author's perspective | Failed to understand the author's perspective in the evaluated paragraph and its connection to the article | The evaluated paragraph is fairly clear in terms of structure, laying the foundation for subsequent analysis by explaining the 4C marketing theory ... (**which is not the author's intention**) | Type I |
| Over-elaboration of non-essential information | Overemphasizing details, reversal of priorities in structures | ((**already presented evidence**) ... further specific evidence is needed to support its conclusion, particularly in clarifying how these strategies hindered technology sharing. | Type II |
| Citation-related error | Incorrect identification of citations or mistaking the citation for the object of evaluation | The evaluated paragraph has logical issues in its argumentation. The author rejects the definition of health based on "bodily integrity ... (**which is the citation part view**) | Type I & II |
| Vague assessment | Copying words from evaluation criteria, with no in-depth revision advice | The argument in this paragraph is relatively clear . However, the supporting evidence appears somewhat limited . And ... | Type II |

Table 3: **Hallucination in LLM essay critiques**: the red background texts are the hallucination part and the **blue**) comments are explanations.

## 4    REDHAT REDUCES HALLUCINATION IN CRITIQUES

### 4.1    BACKGROUND

As we have discussed above, the hallucination in essay critique generation mainly originates from LLMs' un-faithfulness to the essay. LLM is not fully aware of the essay contents, leading to the generation problems when it tries to align with the evaluation criteria.

To bridge the gap between LLM's the faithfulness of the essay and the following of assessing prompt, we propose RedHat. We noticed the phenomenon in education and psychology (Marton & Säaljö, 1976), that breaking the understanding task into question-answering task is able to speed up human's comprehension of long documents. There is an opportunity to ease the evaluation instruction by switching it into series of questions. Then by finding answers from the essay, LLM can reduce its hallucination by more factual information.

### 4.2    CRITERIA EMBODIMENT

Following the idea above, we embody the evaluation criteria $\Gamma$ into a list of questions. To ensure the questions' relevance, we prompted GPT-4 to propose questions conditioned on $\Gamma$ and the essay content. The questions shall cover the essence of $\Gamma$, the above process has to be repetitive to be exhaustive. Formally, denote $\mathcal{E}$ as the essay, $\mathcal{P}$ as the critiquing paragraph, $p_{\text{question}}$ as prompt for this task, questions are produced in the following iterative process:

$$q_n = \text{Question}_\theta(\Gamma, \mathcal{E}, p_{\text{question}}, q_{1:n-1}) \tag{3}$$

The number of questions $n$ is a hyper-parameter. The above process is not economic in reality, with repetition for each new essay exhibiting redundancy on common questions. We repeat the experiments with different essays and pick a list of common questions as the general solution. The questions are reviewed by human writing experts, listed in the Appendix E.

Another important part of criteria decomposition is seeking answers to those questions in the essay. Fortunately, current LLM techniques all showed compelling performance on DocQA and long

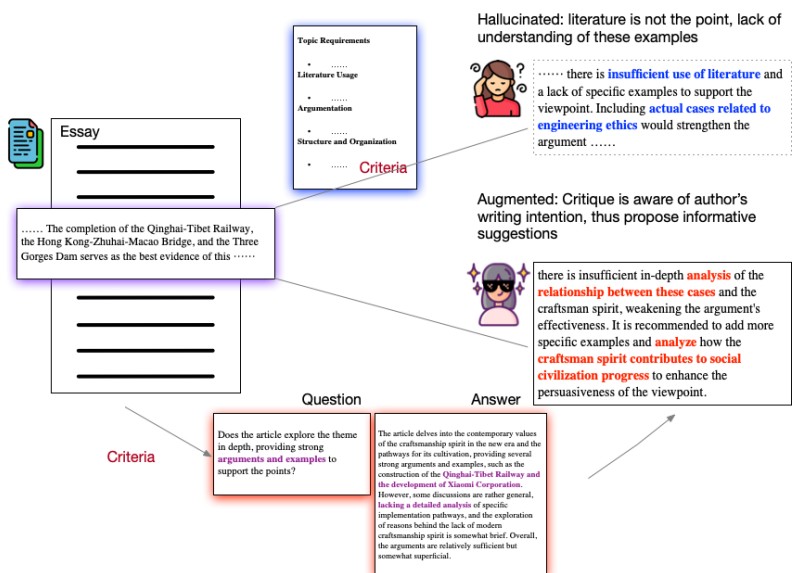

Figure 2: **RedHat Explained**. Converting essay evaluation criteria into a question checklist is beneficial for critique generation. Directly following criteria would ignore the understanding of the essay. RedHats designed to reduce hallucination and ambiguity, and improve critique informativeness. RedHat engages necessary information for understanding the essay in question-answering pairs into the critique generation prompt.

context retrieval (Lewis et al., 2020). The answering process can be streamlined into a separate document question-answering process as formalized below:

$$a_n = \text{DocQA}(q_n, \mathcal{E}) \tag{4}$$

## 4.3 REORGANIZING THE CRITIQUING PROCESS

We state that the direct insert of the question-answering result pairs into the critiquing process is enough to mediate LLM hallucination in critiquing. One of the most practical methods for combining QA results like RAG (Lewis et al., 2020) calls for a domain-specific retriever for such a situation. Though shared with a promising goal to reduce hallucination (Shuster et al., 2021), our task differs from RAG in contents. RedHat tries to rectify LLM's inability to grasp and understand the whole essay, while an essay is presented in the original inputs. RAG supplements external knowledge to relieve the ignorance of LLM knowledge, and usually the knowledge does not explicitly exist in original inputs.

We ground the question-answering results into the original critique generation prompt, as the below formula reveals:

$$c_n = \text{LLM}_{\mathcal{C}}(\Gamma, \mathcal{E}, \mathcal{P}, \{q_i, a_i\}_{i=1}^n, p_{\text{critique}}) \tag{5}$$

## 5 EXPERIMENTS

### 5.1 EXPERIMENT SETUP

**Dataset**: We mainly experiment RedHat on EssayC discussed in section 3. To validate RedHat's effectiveness, we additionally picked a subset from artificial intelligence conference papers as previous works with English-dominated LLMs did. We intentionally chose those papers containing less formulas and illustrations, and more importantly, ensuring the paper authors' are accessible so that they could judge the quality over the generated critiques. We pick 10 paragraphs with longer word counts from each paper to be critiqued. The statistics of the English subset is listed in Table 2.

| | Overall(↑) | Hallu%(↓) | Ambig%(↓) | Info%(↑) |
|---|---|---|---|---|
| Human Critiques | 3.387 | 47.34 | 11.65 | 30.63 |
| Qwen-2-7b-Instruct | 3.187 | 62.53 | 14.68 | 11.90 |
| + 5-shots | 3.178 | 61.01 | 11.14 | 12.66 |
| + RedHat | 3.267 | 62.03 | **7.59** | 15.70 |
| + RedHat-weak | **3.323** | **58.73** | 8.10 | **18.73** |
| + PT | 2.777 | 71.65 | 24.30 | -9.11 |
| + SFT | 2.615 | 74.43 | 27.59 | -19.24 |
| + SFT+RedHat | 2.636 | 77.72 | 22.28 | -10.63 |
| glm-4-9b-chat | 3.190 | 65.99 | 14.97 | 6.60 |
| + RedHat | **3.327** | **63.96** | **9.64** | **13.20** |
| + RedHat-weak | 3.246 | 65.99 | 13.45 | 6.35 |
| + PT | 3.053 | 69.04 | 23.86 | -3.04 |
| + SFT | 2.503 | 79.44 | 16.50 | -31.72 |
| + SFT+RedHat | 2.574 | 79.44 | 14.47 | -27.92 |
| ChatGPT-4o | 2.448 | 76.92 | 11.99 | -9.99 |
| + RedHat | **3.549** | **42.96** | **8.99** | **23.98** |

Table 4: **Main experiment** on EssayC (GLM-4 and Qwen-2) and English subset (ChatGPT-4o). All results are judged by human experts. **Hallu** is short for hallucination (0-100%). **Ambig** is short for ambiguity(0-100%). **Info** is short for informativeness(-100-100%). Due to the discriminating ability of human, the three dimensions are evaluated in human detection of fallacies or goodness. Beside the three dimensions, an **Overall score** is given mainly based on hallucination based on the number of deficits detected.

**Base LLMs**: To validate RedHat's effectiveness in more LLMs, we select GLM-4-9B-chat (GLM, 2024) and Qwen-2-7B-Instruct (Qwen, 2024) to be studied on EssayC. We select ChatGPT-4o to study the English conference paper subset.

**Baselines**: Since there are plenty of human written critiques in EssayC construction, supervised-finetuning (SFT) is a direct baseline method. **SFT** tries to show whether it is applicable to avoid hallucinating from direct learning from teachers' critiques. **Post-pretraining (PT)** tries to clarify our doubt about whether hallucination originates from alienness to long document form reading. **Few-shot** tries to explore the feasibility of bypassing hallucination with in-context examples. Details for few-shot, training and data preparation can be found in Appendix F. Additionally, we also apply RedHat to the supervised finetuned model, to investigate its further application. We are also interested in the quality of answers to the RedHat questions, therefore we compared the LLM self-generated answers in inference (**Weak**) and GPT-4 generated answers.

**Metrics**: Each of the critique is evaluated with four dimensions: hallucination, ambiguity, informativeness and overall. **Hallucination** ($\downarrow 0 \sim 100\%$) is evaluated by the true or false detection rate. If one falls to fit the 6 hallucination types mentioned in Table 3, it is marked as true in hallucination. **Ambiguity** ($\downarrow 0 \sim 100\%$) is calculated whether the critique is ambiguous or not. **Informativeness** ($\uparrow$ -100 $\sim$ 100%) is calculated whether the critique provided useful improvement advice for polishing. They scored three levels of informativeness: of positive help, of no help, of negative help. **Overall score** ($\uparrow 0 \sim 5$) models the task target in Formula 1, and is calculated through: (1) minus 2 per hallucination found; (2) minus 1 for ambiguous; (3) minus 1 for being of negative help or plus 1 for being of positive help (4) truncate into interval 0 to 5.

**Evaluator**: We mainly refer to trained human graduate teaching assistant scores as evaluation results. The details of our human annotations are listed in the Appendix H. We also conducted pairwise preference annotation with base-LLM and RedHat generated critiques. In this scene, human ranks two critiques into which is better or both is good or bad. Each generated comment is annotated by two graduate teaching assistants. In case of discrepancies, a third graduate teaching assistant makes the final decision. Our overall Inter Annotator Agreement is 0.71 in GLM-4 and Qwen-2 as a whole, ensuring annotation consistency and reducing random interference. We also utilized GPT-4o as auto evaluation method to explore the accessibility of automatic evaluators.

## 5.2 MAIN RESULTS

We showed the results in Table 4, with Qwen-2-7B, GLM-4-9B, ChatGPT-4o. Statistics from the `RedHat` (**Orange** background) showed increments in all dimensions compared to base-LLMs. Few-shot benefits the base Qwen2 but is less evident compared to `RedHat`. However, SFT and PT cause decrement in all dimensions, indicating that direct adjust LLM parameters in the aim of fitting MLE loss are not solutions to hallucination reduction in essay critique generation. Additionally, the reduction of hallucination usually correlates to the reduction of ambiguity and the increment of informativeness. Last but not least, considering `RedHat` and `RedHat`-weak, answers provided by GPT

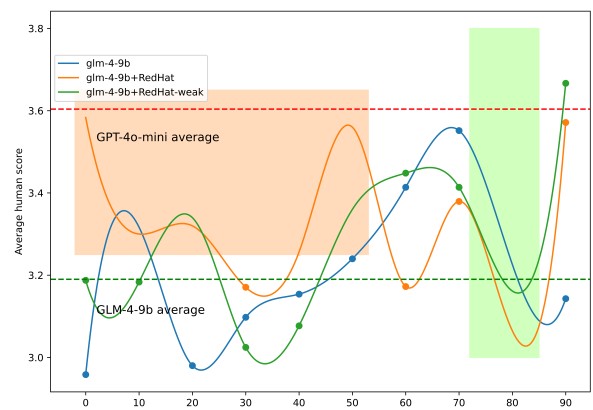

Figure 3: **Distribution** of overall scores with the **position** in the article. The x-axis shows the relative length of the annotated text to the essay. The y-axis shows the average overall score by a human.

(regarded as an DocQA oracle for its high accuracy) or LLM itself all contributed to hallucination reduction and overall improvement.

Figure 3 depicts a dynamic relation between the critiqued piece and its position in the essay. In the **Orange** box, `RedHat` mainly relieved the hallucination in this part. At the 80% point of the article, we observe a notable decline in performance across all methods, as the **Green** box highlights. We hypothesize that this is where the author begins to conclude their argument, rather than continuing to elaborate further. At this stage, the model tends to overextend by providing more detailed explanations than necessary.

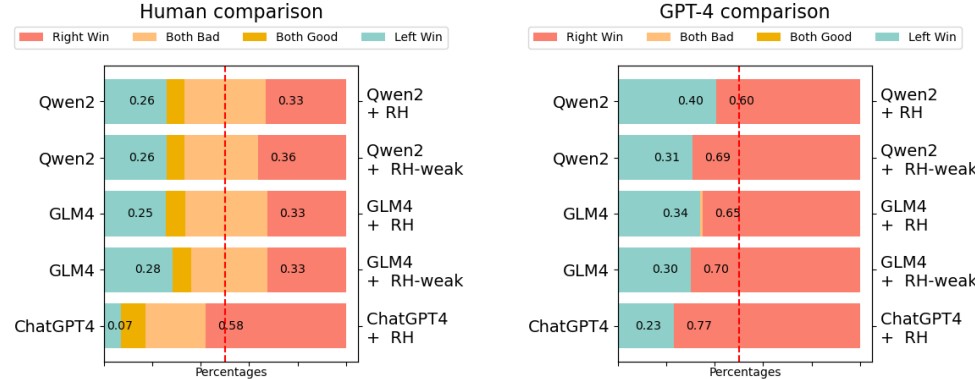

Figure 4: **Results from comparison of critiques generated by baseline with our methods**. Both human experts and GPT-4o judgements are plotted. **RH** is short for `RedHat`.

The comparison between baseline-LLM and `RedHat` are shown in Figure 4. In the figure, human are more preferred to critiques generated by `RedHat` by $\Delta$ 7.11 % in GLM, 10.36% in Qwen. On the one hand, the high tie rates in human judgments result from the number of hallucination types. If one of the six hallucination types is detected from each of the critiques, the pair would be graded as *both is bad*. On the other hand, GPT-4o as comparison evaluator showed low *tie rate*, indicating its potential bias or unawareness of hallucination. Appendix G contains a detailed discussion of them. In conclusion, the overall trend of GPT-4o judgments matches with human judgments and shows the improvements from .

| | Question | | | | Answer | | | |
|---|---|---|---|---|---|---|---|---|
| | R-L | B-1 | BLEURT | BERTScore | R-L | B-1 | BLEURT | BERTScore |
| Qwen-2-7B-Instruct | 10.64 | 15.62 | 27.45 | 71.09 | 11.16 | 6.69 | 21.39 | 74.50 |
| +RedHat | 11.13 | 18.68 | 25.24 | 72.39 | 12.17 | 8.40 | **31.00** | **76.16** |
| +RedHat-weak | 10.92 | 18.95 | 26.24 | 72.50 | 12.21 | 8.69 | 29.41 | 76.41 |
| +SFT | 8.41 | 4.60 | 41.21 | 65.11 | 6.40 | 1.88 | 23.42 | 66.63 |
| +SFT+RedHat | 8.73 | 5.10 | 44.56 | 65.50 | 6.75 | 2.03 | 21.55 | 67.22 |
| +PT | 8.20 | 4.07 | 40.82 | 64.81 | 6.05 | 1.67 | 27.49 | 66.02 |
| GLM-4-9B-chat | 8.89 | 5.69 | 42.00 | 66.09 | 9.88 | 3.55 | 55.28 | 68.00 |
| +RedHat | 9.88 | 7.51 | 44.26 | 67.31 | 10.18 | 4.43 | 55.51 | 69.72 |
| +RedHat-weak | 9.96 | 7.54 | 44.11 | 67.17 | 10.52 | 4.60 | 55.71 | 69.90 |
| +SFT | 7.90 | 3.75 | 37.45 | 64.25 | 6.75 | 2.16 | 50.36 | 65.04 |
| +SFT+RedHat | 8.28 | 4.28 | 40.08 | 64.85 | 7.25 | 2.42 | 51.32 | 65.96 |
| +PT | 8.73 | 5.77 | 41.10 | 65.90 | 8.91 | 3.54 | 54.07 | 67.89 |

Table 5: **Overlaps between generated critiques and questions**. **R-L** is short for Rouge score calculated with longest common substrings. **B-1** is short for BLEU score calculated with unigrams.

## 5.3 HOW QAS HELP REDUCE HALLUCINATION?

To explore how QA results assist in comment generation, we designed the following analytical experiments to investigate the impact of QA accuracy on outcomes and the overlap between the generated critiques and the QA.

**Question-Answer Quality**: We evaluate the validity of questions by analyzing the similarities between critiques, questions and answers. We calculated word-level overlapping with **ROUGE** (Lin, 2004) and **BLEU** (Papineni et al., 2002), and semantic similarity with **BLEURT** (Sellam et al., 2020), **BERTScore** (Zhang et al., 2019), between the generated critiques and the corresponding questions list, answers list, as shown in Table 5.

We can observe several findings from the results in Table 5. First, with RedHat, similarities between generated critiques and questions do not significantly increase, indicating the questions are not leaking the desired contents to the LLM. Second, similarities gain with answers is observed, especially with Qwen + RedHat, showing that detailed information about the essay is conveyed in the answers by RedHat.

**Answer Accuracy Influence on Performance**: In our methodology, we assume that the responses to the questions are correct, which are generated by a perfect long-document question-answering model. We invited human essay evaluation experts to score the correctness of answers for the questions on different essays. GLM-4 show a 14.4% error rate, followed by 7.8% from Qwen-2 and 4.4% ChatGPT-4o. The decrease in error rates corresponds to the gain in point-wise scoring (RedHat-weak rows) of hallucination and

| | GLM-4 v.s. GLM-4-RedHat | | | |
|---|---|---|---|---|
| | Win | Tie | Lose | Δ |
| Human | 45.74 | 5.32 | 48.94 | 3.20 |
| GPT-4o | 19.56 | 58.67 | 21.78 | 2.22 |

Table 6: **Critique Effect on Essay Polish**. Preference picking between through human and GPT-4o-0815.

pairwise comparison. However, the overall influence of RedHat still outperform baseline-LLMs, suggesting the robustness of our method.

## 5.4 HOW UPDATED CRITIQUES HELP WITH ESSAY POLISH?

Comment generation needs to serve the educational context, meaning it should assist students in improving their essay quality. We designed an experiment to verify that the comments generated by the RedHat can better help improve text quality.

We selected 100 essay samples from the test set and paired the comments generated by glm-4-9b-chat and glm-4-9b-chat-RedHat. We used GPT-4o as a professional text enhancer, prompting it to

| | glm-4-9b-chat | glm-4-9b-chat+`RedHat` |
|---|---|---|
| Paragraph | The renovation of Juer Hutong involves two key aspects: one is addressing planning issues from the perspective of a historically and culturally significant city, and the other is dealing with residential design from the standpoint of urban design. In terms of historical and cultural city planning, there are two main issues: first, the facade is inconsistent. The upper floors of No. 41 Juer Hutong are white, which appears out of place against the primarily gray hutong landscape. Second, multi-story buildings still disrupt the traditional hutong atmosphere to some extent. Although the three-story buildings in No. 41 maintain the traditional courtyard-style rooftops, preserving the urban texture from an aerial view, from within the hutong, the height of the buildings blocks the view, offering an experience similar to that of high-rise residential complexes. | |
| Critique | This paragraph has issues with its structural organization. The main argument at the beginning is not entirely consistent with the discussion in the latter part, which disrupts the logical flow of the text. It is recommended to revise "The renovation of Juer Hutong involves two key aspects" to "There are two issues in the renovation of Juer Hutong: first, the planning of the historical and culturally significant city, and second, the urban design." | This paragraph reveals shortcomings in structural organization. It analyzes the issues of the Juer Hutong renovation from the perspective of a historically and culturally significant city but abruptly shifts to the analysis of a specific case (No. 41 Juer Hutong) without proper transition or explanation. It is recommended to briefly outline the main argument after mentioning the planning issues of the historical and culturally significant city, then proceed to the detailed analysis with references, in order to enhance the internal logic and overall coherence. |
| Overall | 2.0 | 5.0 |
| Reasons | Over-correction at the word or sentence level; of no use | provided informative suggestions |

Table 7: Example between the quality of two critiques by glm-4-9b-chat and glm-4-9b-chat + `RedHat`. Already translated into English from the original Chinese texts.

revise the annotated text based on the comments. Finally, we invited master and doctoral students, as well as teachers from the humanities, to compare the quality of the revised texts. The results are shown in the Table 6. From the results, it is evident that our method, using GPT-4o as the enhancer, can effectively generate in-depth comments and improve text quality. Also, we find that there is a huge gap between human evaluation and GPT-based automatic evaluation. Which is another hint on the unexposed bias inside LLMs.

## 5.5 CASE STUDY

We pick one case from a certain student essay titled "Why was the renovation of Ju'er Hutong highly praised but not widely promoted". In the specific paragraph presented, the author made an abrupt twist in conveying from "the two aspects of renovation" to "two main issues of historical and cultural city planning". Although glm-4-9b-chat points out the structural issues in it, it mistakenly focuses on the revision of the terms. On the other hand, glm-4-9b-chat with `RedHat` recognized the issue and provided suggestions to outline the main argument. This benefits the coherence both in the paragraph and the essay, showing the augmentation of the LLM's ability against hallucination caused by local phrases.

## 6 CONCLUSIONS

In this work, we proposed `RedHat`, an effective method for reducing hallucinaiton in LLM-generated critiques in essay assessment. `RedHat` enhanced GLM-9b-chat, Qwen-2-7B-Instruct and ChatGPT-4o by adding an essay-level digest in a question-answering format for the LLM. In our pedagogical application setting, results showed that our method reduced hallucination, ambiguity and improved their informativeness. On the other hand, our generated critiques also greatly helped polish the original essay content. The method is both effective in reducing the hallucination both with `EssayC` and with the English conference papers.

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

# A   EXTENDED DISCUSSION OF RELATED WORK

## A.1   HALLUCINATION DATASOURCES

The halluciniation in natural language generation task is generally defined as the phenomenon that model generated contents contain information that contradicts or is unfaithful to user instructions, additional background context, and/or factual knowledge. Various previous studies have attempted to mitigate hallucination problem in enormous traditional NLG tasks. Due to their discrepancy in task formats, nevertheless, these works define hallucination in rather different ways and aspects and design methodologies tailored to solving these problems in concern. In conversation tasks, (Zhang et al., 2018) proposed PersonaChat dataset aiming to relieve the problem of self-consistency violation in chit-chat. (Dinan et al., 2019) attempts to incorporate external knowledge corpus for more factual knowledge-based dialogue generation. In abstractive summarization tasks (a most related domain of our task), efforts have been paid to alleviate the hallucination problems embodied as generating spans not entailed by the source text. Early works explores methods to improve factuality from source content understanding (Huang et al., 2020), training process (Cao & Wang, 2021) and post-training phase (Dong et al., 2020).

## A.2   HALLUCINATION DETECTION

Such method needs external knowledge sources, or reference answer for judging. There are also pioneers who invented reference free methods. FEWL(Wei et al., 2024) weights multiple LLMs answers as proxy of golden answers, which theoretically provided plausibility for judgment. (Hou et al., 2024) utilizes the belief of LLM to check their hallucination problem via decomposing statements into child statements to check in a hierarchical way.

Essay evaluation is both reference and knowledge sparse task, making it hard for quantification on judging. Our method inherits the above ideology by embodying the concept of faithfulness to essay as correctly performs the docOA task.

## A.3   HALLUCINATION CAUSES

The causes of hallucination on knowledge-intensive tasks are various. Previous works have focused on those arising from deficiencies in data collection and preprocessing, training, and inferencing phases. In terms of data sources, the emergence of hallucination could be attributed to incorrect or biased data, absence of real-time or proprietary knowledge, or wrong utilization of knowledge (Huang et al., 2023). In the training phase, (Sharma et al., 2023) shows that the training process of RLHF may wrongly lead LLM to produce content that flatter users but disobeys facts. In the inference phase, (Wang & Sennrich, 2020) claims that the discrepancy between the training and inferencing pattern of the AR model could lead to hallucination. (Zhang et al., 2023) finds that hallucinations already generated can mislead LLM to continue producing error statements.

In our work concerning hallucination in essay evaluation tasks, hallucination could be caused by more complex factors. Due to blurred or even seemingly contradictory criteria of judgment, evaluators could generate outputs not consistent with previous contents, likewise tending to generate tangential evaluations.

## A.4   ESSAY CRITIQUE GENERATION

Utilizing LLMs to judge and refine human writing has become a buoyant application of recent LLM systems. Several systematic evaluations have been conducted on the capability of LLM to generate critique content for human writings in various scenarios (Tang et al., 2024; Rahman et al., 2023; Lagakis & Demetriadis, 2021; Jong et al., 2023; Lagakis & Demetriadis, 2021). There are also emerging systems built for providing critique generation (Tyser et al., 2024; Peng et al., 2024; Gong et al., 2021; Kumar et al., 2021; Li et al., 2021), manifesting remarkable performance. The primary difference between their work and ours is that their system focuses on generating evaluative comments, whereas we prioritize minimizing hallucinations in the feedback to help writers improve text quality. Also, there lacks of an agreement on a unified testbench.

# B LIMITATIONS

There are two limitations of this work. First, the development of automated hallucination detection techniques for essay critique generation is necessary but requires extensive data labeling, which was constrained by practical budget limitations; thus, we believe it is important to explore synthetic data for the purpose as a focus for future research. Second, exploring how LLM-generated critiques influence LLM-generated essays could deepen our understanding of LLM-based automatic reviews. If successful, it will greatly improve the potential of LLMs for enhancing human-written texts.

# C CRITERIA FOR ESSAYS

## C.1 CHINESE ARGUMENTATIVE ESSAYS

The essay content studied in our work exhibits four structural and content characteristics. **Topic** is the background and the author's core perspective to be delivered in the essay. An essay must have a well-defined topic to discuss. Students need to choose a focused, valuable question from a clearly identified discipline that allows for in-depth discussion. **Literature** is the bridge between the essay and the information outside the essay. It is essential to engage in a thorough discussion about existing literature to clearly understand the issue at hand and cite sources appropriately throughout. **Arguments and Structure** refer to the chain of thoughts that depict how arguments are articulated. When presenting arguments, the structure should follow the "problem-argument-reason-evidence" structure to ensure persuasiveness. Arguments should be clear, well-supported, and employ proper logical reasoning, often utilizing both deductive and inductive methods. **Language and Norms**: First-person pronouns should be avoided, and the arguments must be original. When referring to others' opinions, it is crucial to paraphrase appropriately and refrain from plagiarism.

Student essays are evaluated with respect to the standards uniformly above. We believe such criteria are beneficial for narrowing down possible variances stemming from different assessors' subjectivity. When evaluating the model's generated critiques, human labelers can then better focus on hallucinations in the critiques, conditioned on the above criteria.

## C.2 ENGLISH CONFERENCE PAPERS

Generally, we refer to ICLR 2025 review instructions for details (https://iclr.cc/Conferences/2025/ReviewerGuide). We applied the ICLR reviewer guidelines as evaluation criteria. Since ICLR reviewer guidelines have already contained more than 10 questions in it, we replace the guideline questions with description of the expectation for a good conference paper on those questions. The following is the evaluation criteria version without questions we used.

```
1. Thoroughly Read the Paper: The paper should be read carefully
in its entirety. Related works and citations must be reviewed to
ensure a comprehensive evaluation. Sufficient time should be
allocated for this process.

2. Key Considerations While Reading:

2.1 Objective of the Work: The paper should have a clear goal,
such as addressing a known problem or application, highlighting a
new issue, or presenting new theoretical findings. Different
objectives should be assessed based on their potential value and
impact.

2.2 Strong Points: The submission should be clear, technically
correct, experimentally rigorous, reproducible, and present novel
findings in areas such as theory or algorithms.

2.3 Weak Points: Any shortcomings in clarity, technical
correctness, rigor, reproducibility, or novelty should be noted.
```

2.4 Open-Mindedness: The value of the paper should be considered from the perspective of the entire ICLR community, even if it may not seem immediately relevant or interesting to individual reviewers.

3. Evaluating Core Aspects for Recommendation:

3.1 Problem Definition: The paper should tackle a specific question or problem with clarity.

3.2 Motivation and Context: The approach should be well-motivated and appropriately contextualized within the literature.

3.3 Support for Claims: The paper should provide rigorous evidence to support its claims, ensuring results are both correct and scientifically valid.

3.4 Significance: The work should contribute new, valuable knowledge to the community, whether empirical, theoretical, or practical, regardless of whether it achieves state-of-the-art results.

4. Initial Review Structure:

4.1 Summary: Clearly summarize the paper's contributions in a positive and constructive manner.

4.2 Strengths and Weaknesses: Identify the paper's strong and weak points comprehensively.

4.3 Initial Recommendation: Provide an initial recommendation (accept or reject) with a clear rationale.

4.4 Supporting Arguments: Present evidence and arguments that support the recommendation.

4.5 Clarifying Questions: Include questions for the authors to address ambiguities and provide additional evidence for the assessment.

4.6 Improvement Suggestions: Offer constructive feedback aimed at improving the paper. Clarify that these suggestions are for improvement and not necessarily decision-critical.

5. Complete the CoE report:

5.1 Familiarize yourself with the ICLR Code of Ethics before starting reviews.

5.2 Assess whether the paper has potential CoE violations and provide explanations if applicable. The CoE report will involve answering these questions as part of the review process.

6. Active Participation in Discussions:

Actively engage in the asynchronous discussion phase, where reviewers, authors, and area chairs exchange feedback. Be open to revising your initial recommendation based on new insights or updates to the submission.

```
7. Borderline Paper Discussions:

Participate in virtual meetings organized by Area Chairs (ACs) to
discuss borderline cases. Familiarize yourself with feedback from
other reviewers to contribute meaningfully to the discussions.
Reviewers who fail to attend without emergencies will have their
absence noted.

8. Final Recommendation:

Update your review to reflect any new information or revisions
during the discussion phase. Clearly articulate the reasoning
behind your final recommendation, including what influenced any
changes to your assessment.
```

With the above criteria, the prompts for English conference paper critiquing is structured as followed.

```
Suppose you are a professional essay polisher for international
conference in learning representation. Based on the following
review criteria, provide suggestions to improve the appointed
paragraph.

[review criteria begins]
{criteria}
[review criteria ends]

[paper begins]
{paper}
[paper ends]

[paragraph begins]
{paragraph}
[paragraph ends]

Now begin your suggestions within 100 words. Your suggestions
should aim at pointing out the weaknesses and providing
constructive feedback.
```

## D  DATA PREPARATION

We collect over 6,000 student essays from our course archives from Fall 2019 to Spring 2024, and randomly select 50 essays to serve as the test set for our evaluation.

Below are our considerations for picking:

**Diversity of Topics**: The selected 50 essays cover a broad spectrum of topics, including literature, cultural criticism, gaming industry reviews, electric vehicles, technology, and artificial intelligence. These topics were categorized into distinct thematic groups to ensure a diverse representation of subject matter for our testing.

**Content and Instructor Feedback**: All essays were initial drafts submitted by students for one-on-one feedback from their course instructors. The instructors provided paragraph-level comments, primarily focusing on the writing issues and offering suggestions for improvements.

**Ethics and Privacy Considerations**: To ensure the ethical use of student data, we obtained approval from the course teaching team for the use of these essays. Additionally, all essays were anonymized

by removing personal identifiers such as student names, IDs, locations, and any other sensitive information. We applied standard anonymization techniques to ensure privacy and a manual review was conducted to confirm that no personal information remained in the dataset.

**De-noising**: We apply format-revision and correction to the the essays. We also filter out very casual teacher comments like single punctuation like '?', or commenting on unrecognizable pieces.

# E  ESSAY READING QUESTION LIST

## E.1  QUESTIONS FOR CHINESE ARGUMENTATIVE ESSAYS

We list the questions in Table 8 that we collected from the essay writing experts. They are crucial questions in understanding an essay. The picking threshold is the agreement over 15 TAs and instructors.

| Setting | Prompt |
| --- | --- |
| **Question 1** | 文章是否有一个明确的主题或中心思想? 
 Does the article have a clear theme or central idea? |
| **Question 2** | 作者在文章的开头是否清晰地提出了主要观点或论点? 
 Does the author clearly present the main point or argument at the beginning of the article? |
| **Question 3** | 作者是否清晰地表达了他们的观点, 且这些观点在文章的各部分中得到一致的支持和阐述? 
 Has the author articulated their views clearly, with consistent support and elaboration throughout the various sections of the article? |
| **Question 4** | 这些观点是否贯穿全文, 有没有与主题无关的内容? 
 Are these viewpoints consistently maintained throughout the text, or is there unrelated content? |
| **Question 5** | 文章是否深入探讨了主题, 提供了有力的论据和例子来支持观点? 
 Does the article delve deeply into the subject, providing strong evidence and examples to support its arguments? |
| **Question 6** | 作者是否展示了对题目有深刻的理解和分析, 还是仅仅停留在表面? 
 Does the author demonstrate a profound understanding and analysis of the topic, or do they merely scratch the surface? |
| **Question 7** | 作者在文章中是否深入分析了主题, 提供了充分的论据、例子和细节来支持他们的观点? 
 Has the author thoroughly analyzed the theme within the article, offering ample evidence, examples, and details to back up their points? |
| **Question 8** | 有没有考虑到不同的视角或反驳意见, 并且对这些进行了回应? 
 Have different perspectives or counterarguments been considered, and have these been adequately addressed? |
| **Question 9** | 文章中的语言是否清晰、准确且具有表现力? 
 Are the statements in the article clear, accurate, and expressive? |
| **Question 10** | 语言风格是否与文章的目的和受众相匹配? 
 Does the writing style align with the article's purpose and audience? |
| **Question 11** | 文章是否有明显的语法、拼写或标点错误? 这些错误是否会干扰读者的理解或降低文章的专业性和可信度? 
 Are there noticeable grammatical, spelling, or punctuation errors in the article? Do these errors hinder the reader's understanding or diminish the professionalism and credibility of the piece? |
| **Question 12** | 文章是否提出了独特的见解或创新的观点, 或者只是重复了常见的观点? 
 Does the article present unique insights or innovative viewpoints, or does it merely reiterate common ideas? |
| **Question 13** | 有没有引入新颖的例子或视角来讨论主题, 从而使文章在众多类似文章中脱颖而出? 
 Has the author introduced novel examples or perspectives to discuss the theme, allowing the article to stand out among similar works? |
| **Question 14** | 文章的结构是否合理? 
 Is the structure of the article logical? |
| **Question 15** | 段落之间的衔接是否流畅? 
 Is there a smooth transition between paragraphs? |
| **Question 16** | 作者是否按照一个清晰的逻辑顺序来组织他们的论点和证据? 
 Does the author organize their arguments and evidence in a clear logical sequence? |
| **Question 17** | 每一段是否都有一个明确的中心思想, 并且与前后的段落自然衔接? 
 Does each paragraph have a distinct central idea that connects naturally with the preceding and following paragraphs? |
| **Question 18** | 段落之间是否有过渡句来帮助读者理解文章的整体结构? 
 Are there transitional sentences between paragraphs to assist the reader in understanding the overall structure of the article? |

Table 8: Crucial questions list for EssayC.

## E.2 Questions for English Artificial Intelligence Conference Papers

We select the question list from the ICLR guideline and list them in Table 9.

| Setting | Prompt |
| --- | --- |
| **Question 1** | What is the goal of the paper? |
| **Question 2** | Is it to better address a known application or problem, draw attention to a new application or problem, or to introduce and/or explain a new theoretical finding? A combination of these? |
| **Question 3** | Is the submission clear, technically correct, experimentally rigorous, reproducible, does it present novel findings (e.g. theoretically, algorithmically, etc.)? |
| **Question 4** | What is the specific question and/or problem tackled by the paper? |
| **Question 5** | Is the approach well motivated, including being well-placed in the literature? |
| **Question 6** | Does the paper support the claims? |
| **Question 7** | Are results, whether theoretical or empirical, correct and scientifically rigorous? |
| **Question 8** | What is the significance of the work? |
| **Question 9** | Does it contribute new knowledge and sufficient value to the community? |
| **Question 10** | Does the paper convincingly demonstrate new, relevant, impactful knowledge (including empirical, theoretical, for practitioners, etc.)? |
| **Question 11** | What questions would you like answered by the authors to help you clarify your understanding of the paper and provide the additional evidence you need to be confident in your assessment? |
| **Question 12** | Is there a potential violation of the Code of Ethics (CoE)? |
| **Question 13** | If there is a potential violation, why might there be a potential violation? |

Table 9: Crucial questions list For English artificial intelligence conference papers.

# F Details for training and implementations

## F.1 Supervised finetuning

### F.1.1 Data Preparing

We conducted our supervised finetuning over augmentation of teachers' original comments from historical archives apart from the test set. We found original teachers' comments are informal and fragmented, and directly finetuning on them causes damage to the LLM's performance. Therefore, we extracted teachers' comments and deployed a GLM-4-130B for augmentation. The aim of augmentation is to rewrite the semantically low-quality comments into fluent ones, easing for LLM to fit on. The prompt for augmentation can be found in Appendix I.

As a result, we adopt 31,694 polished human paragraph-level critiques as training data, excluded from the `EssayC` testset split mentioned in Section 2. The format of the data is arranged into (evaluation prompt, essay, and target paragraph) as input, and polished paragraph as output. The train and valid set are split based on essays to avoid potential leakage.

### F.1.2 Training Details

We split the data into training and validation sets with a 0.95:0.05 ratio. The training epoch is set as 1.15, for from empirical observation, the lowest loss on the validation set falls around epoch 1.1 to 1.2. We adjust learning rate from {1e-5, 2e-5, 3e-5, 5e-5, 1e-4}, weight decay rate {1e-3, 1e-2}, betas for Adam {[0.9, 0.999], [0.9, 0.9]}, scheduler between {linear, cosine}. Finally, we pick the following config for the least evaluation loss. The training is implemented with LLaMA-Facotory (Zheng et al., 2024b).

- per_device_train_batch_size: 1
- gradient_accumulation_steps: 2
- learning_rate: 1.0e-5
- weight_decay: 0.01

- adam_beta1: 0.9
- adam_beta2: 0.999
- max_grad_norm: 1.0
- num_train_epochs: 1.15
- lr_scheduler_type: cosine
- warmup_ratio: 0.1

## F.2 POST TRAINING

### F.2.1 DATA PREPARING

As for post-pretraining, we follow two steps: (1) pre-training on Chinese academic papers in the field of literature, social science and humanities and (2) followed by SFT on the previous data to ensure the alignment of the critiquing task.

We crawled 128,321 academic papers from the **Chinese National Social Science Base**. The papers mainly come from journals, such as *Exploration and Free Views, Fiction Monthly Shanghai Literature, Beijing Literature Novella Month, Science Technology Critiques, Tanzhen Technology Review* and so on. We use OCR with doc2x API (https://v2.doc2x.noedgeai.com) and applied the follow-up data filter and typo fixing with GPT-4 and GLM-4. The above process produces 27,430 pure text papers of an average around 30,000 Chinese characters. The whole tokens surpassed 1.5 billion.

### F.2.2 TRAINING DETAILS

We split the data into training and validation sets with a 0.95:0.05 ratio. The training epoch is set as 6.0, for from empirical observation, the lowest loss on the validation set falls around epoch 5.0 to 7.0.

We adjust learning rate from {1e-5, 2e-5, 3e-5, 5e-5, 1e-4}, weight decay rate {1e-3, 1e-2}, betas for Adam {[0.9, 0.999], [0.9, 0.9]}, {linear, cosine}. Finally, we pick the following config for the least evaluation loss. The training is implemented with LLaMA-Facotory (Zheng et al., 2024b).

- per_device_train_batch_size: 1
- gradient_accumulation_steps: 1
- learning_rate: 3.0e-5
- weight_decay: 0.01
- adam_beta1: 0.9
- adam_beta2: 0.999
- max_grad_norm: 1.0
- lr_scheduler_type: cosine
- warmup_ratio: 0.1
- bf16: true

## F.3 FEW-SHOT IMPLEMENTATION

In our experiment, we experimented with 5-shot structure to test its feasibility to handle the task. The structure of 5-shot is listed as follows. Note that the beginning of the prompts and the ending of the prompts remain the same as prompts for baseline-LLM inference in Table 10. The only change is the insertion of the five examples.

```
[Evaluation Prompt begins and ends]

Explanation of the criteria.
```

[Evaluation Criteria begins and ends]

Explanation of the essay.

[Target essay begins and ends]

Explanation of paragraph.

[Target Paragraph begins and ends]

There are five examples for your critiques. You can refer to them or mimic.

[Example 1 begins]
Target Essay 1
Paragraph 1
Critique 1
[Example 1 ends]

[Example 2 begins]
Target Essay 2
Paragraph 2
Critique 2
[Example 2 ends]

[Example 3 begins]
Target Essay 3
Paragraph 3
Critique 3
[Example 3 ends]

[Example 4 begins]
Target Essay 4
Paragraph 4
Critique 4
[Example 4 ends]

[Example 5 begins]
Target Essay 5
Paragraph 5
Critique 5
[Example 5 ends]

Now, please provide your evaluation. Note that although five aspects are listed in the evaluation criteria, you only need to evaluate one dimension based on the most prominent feature in the paragraph. In your evaluation, please integrate your notes to grasp the overall framework, thought process, and logic of the article. Your feedback should help the student improve the quality of the paragraph. If there are issues, please point them out and offer suggestions for improvement. Please respond with your feedback directly without using formalities, and your evaluation should not exceed 100 word.

|        | Mode    | Win  | Tie  | Lose |
|--------|---------|------|------|------|
| Pair A | Forward | 0.16 | 0.01 | 0.82 |
|        | Reverse | 0.19 | 0.02 | 0.80 |
|        | Average | 0.18 | 0.02 | 0.81 |
| Pair B | Forward | 0.24 | 0.01 | 0.76 |
|        | Reverse | 0.45 | 0.01 | 0.54 |
|        | Average | 0.34 | 0.01 | 0.65 |
| Pair C | Forward | 0.68 | 0.00 | 0.32 |
|        | Reverse | 0.80 | 0.00 | 0.20 |
|        | Average | 0.74 | 0.00 | 0.26 |
| Pair D | Forward | 0.46 | 0.02 | 0.52 |
|        | Reverse | 0.63 | 0.01 | 0.36 |
|        | Average | 0.55 | 0.01 | 0.44 |

Figure 5: **Position Bias** by GPT evaluator. **Forward** shows that critique A is posited far from the end of the prompt while **Reverse** is the opposite case. The scores we reported are the algorithmic average of the two modes.

# G    POSITION BIAS OF THE EVALUATOR

We observe significant position bias on the pairwise scoring of GPT-4o-mini. As we find in Table 5. We compared four settings from top to down:

- GPT-4o-mini-0718 V.S. GPT-4o-mini-0718-`RedHat`
- glm-4-9b-chat V.S. glm-4-9b-chat-`RedHat`
- glm-4-9b-chat V.S. glm-4-9b-chat-`RedHat`-weak
- glm-4-9b-chat-sft V.S. glm-4-9b-chat-sft-`RedHat`

As the table showed, GPT showed a significant preference on the item that is near to the end of the prompt (**Revsere**). Previous works in multiple choices (Zheng et al., 2023) also discussed such a phenomenon.

# H    HUMAN ANNOTATION

## H.1    WRITING EXPERT INFORMATION

We hired 15 writing experts for the human annotation stage. They are serving as teaching assistants in the undergraduate writing course. The group primarily consists of graduate students and advanced undergraduates (juniors and seniors), representing a diverse range of academic departments. This interdisciplinary composition ensures the accessibility and relevance of articles across various disciplines and research topics.

## H.2    ANNOTATION GUIDELINE TRANSLATED

The following verbatim is our annotation document for human expert annotators. The original document is in Chinese and we translate it into English.

```
Evaluation Scoring and Annotation Guidelines Document (For
criitque quality evaluation)

I. Task Description & Objectives
```

The model is tasked with evaluating human-written paragraphs. However, due to limitations in the model's capabilities, the evaluation may produce instances of hallucination and other issues. The core objective of this task is to assess the overall quality of the model's comments based on specific dimensions and to conduct preference scoring and comparison.

In a given essay, multiple comments are provided for a particular paragraph. Our tasks are as follows:

1. Scoring { Evaluate the quality of comments based on three dimensions (hallucination, detail, and informativeness) and assign scores accordingly.

2. Subjective Ranking { Subjectively rank the quality of selected pairs of comments.

II. Data Field Description

Fixed Fields

- Original Text: The original essay is in document format, which can be accessed for viewing (annotations from the instructor can be seen after downloading).

- Original Paragraph: The paragraph being evaluated by the model, sourced from a specific section of the paper.

- Comments A | H | C | G | I | D | E | F: Eight different model comments on the original paragraph, including opinions on structure, content, and format.

Annotation Fields

- Scores for Comments A | H | C | G | I | D | E | F: Score + corresponding deduction reasons (drop-down list) + 4 sets of preference comparisons, totaling 20 points.

1. Comment Scoring (8 scores + corresponding multiple-choice reason boxes):

   - Each comment is scored out of a maximum of 5 points, with deductions made based on error types; specific rules can be found in STEP 3.

2. Preference Selection (4 single choices):

   - A & H Comment Comparison: Preference comparison between comments A and H.

   - C & G Comment Comparison: Preference comparison between comments C and G.

   - C & I Comment Comparison: Preference comparison between comments C and I.

   - D & E Comment Comparison: Preference comparison between comments D and E.

Preliminary Notes:

The order in the multi-dimensional table from left to right will follow the sequence:

A, H, C, G, I, D, E, F. Reading from left to right generally does not require looking back. Note that preference comparisons will be interspersed throughout.

Scoring is supported by objective dimensions, but these dimensions may not always correspond directly to the actual quality of the comments. Preference selection can include subjective factors, allowing evaluators to choose the most helpful comment between two options.

III. Specific Scoring Rules (Deduction System)

Scores will be assigned based on the following three dimensions, with a total score of 5 points, deducting down to 0 points. If the final score is 5 (full score) and there are no other deduction points, please check the box for constructive feedback (add 1 point) to provide a reason for the full score.

Dimension 1: Hallucination

- A single hallucination error results in a deduction of 2 points, two errors lead to a 3-point deduction, and more than two errors lead to a 4-point deduction. The following rules were previously detailed in the pre-annotation documentation regarding hallucination classification:

1. Ignoring Context and Multimodal Information

    - Explanation: While the entire paper may not provide this information, it can be inferred from the feedback given by human authors whether the model's comments overlook contextual text information or multimodal information (such as images or links).

    - Typical Context Issues: The author may have presented a viewpoint or concept in the surrounding context that the model fails to recognize. This is easily identified with human feedback, but without it, relevant contextual information must be judged.

    - Multimodal: When the model evaluates articles that combine text and images, it may fail to effectively parse and integrate the meanings of the illustrations within the text, leading to deviations or errors in assessing the relationships between text and images.

2. Vocabulary, Grammar, and Punctuation Correction Hallucinations (Overcorrection, Errors)

    - The model may provide unnecessary overcorrections regarding ordinary vocabulary and grammar in the paper|for example, demanding an explanation for a simple word and providing examples.

– Corrections made to punctuation and grammar may be incorrect.

– Sentences that lack fluency should be categorized in this group.

3. Misunderstanding Concepts, Viewpoints, and Logical Structures

– Failure to recognize or understand the main viewpoints, concepts, and logical structures expressed by the author in the paragraph, yet proceeding to make corrections.

4. Content Structure – Overcorrection of Non-Key Information

– Requires thorough reading and understanding of the original paragraph's theme and arguments, assessing whether the model displays the following issues:

1. Failure to correctly identify the main argument of the paragraph, resulting in corrections that do not align with the actual situation.

2. Proposing expansions or corrections that focus on non-essential information.

3. Errors in summarizing the author's viewpoint.

4. Misunderstanding of the inter-paragraph relationships at the chapter level.

5. Proposing additions or expansions due to a failure to differentiate between the author's argumentation logic and specific concepts.

5. Citation-Related Errors|Content Formatting Comments

– The model may encounter the following hallucinatory issues regarding citations in the paper:

1. Incorrectly treating a citation as an evaluation target.

2. Failing to recognize or incorrectly identifying citation information.

3. Guiding errors in citation formatting.

4. Incorrectly assuming that there is citation information when the original text does not provide any.

Dimension 2: Detail Level

– Deductions of 1 point will be applied for vague evaluations.

– Vague evaluations:

– Comments provided by the model are overly generic and lack substantial content, making them applicable in any context.

Dimension 3: Constructiveness

- Constructive feedback adds 1 point; lack of substantial help
results in a deduction of 1 point.

- Note: If a comment has no issues and is constructive, it can
still receive a score of 5.

- Evaluation lacking helpfulness:

  - The model's comments do not offer constructive suggestions
  that would aid in improving the paper, resulting in a deduction
  of 1 point.

  - It is important to distinguish the constructiveness dimension
  from the hallucination dimension: having hallucinations does not
  automatically warrant a deduction for constructiveness. If the
  AI provides helpful suggestions for improving the paper, then no
  deduction is necessary; however, if the AI misleads the reader,
  then a deduction should be applied.

  - Care should be taken to avoid double deductions stemming from
  hallucination issues that lead to a lack of helpfulness.

- If the comments provided by the model are highly beneficial for
the improvement of the paper, an additional point can be awarded
based on this dimension.

The following is the document for preference picking on polished essays.

I. Task Description & Objectives

People can polish articles of varying quality by following
different types of comments. The core of this task is to score
preferences of the polished text according to specific dimensions
based on the comments.

II. Data Field Description

Fixed Fields

- Original Text: The original essay is in document format, which
can be accessed for viewing (annotations from the instructor can
be seen after downloading).

- Original Paragraph: The paragraph being evaluated by the model,
sourced from a specific section of the paper.

- Polishing A | H | C | G | I | D | E | F: The polished based on
the original text and original paragraph, which are to be
evaluated.

Annotation Fields

1. Preference Selection (3 single choices, win / lose / good tie /
bad tie):

    - A & H Polishing Comparison: Preference comparison between
    polishings A and H.

– C & G Polishing Comparison: Preference comparison between polishings C and G.

– A & J Polishing Comparison: Preference comparison between polishings A and J.

2. Selection Reasons for Preference (choose from 1–5. Please refer to Section III for detailed information.)

III. Criteria for Preference Selection

The following describes the characteristics of high-quality polishing:

1. Adaptability to the Original Text (Original Structure):
   – When the polished paragraph is inserted into the article, does it align with the main flow of the original text, without deviating in the logical chain?
   – The viewpoint of the polished paragraph should not contradict any content already present in the original text.

2. Language Characteristics:
   – Does it comply with the writing norms taught in our writing courses?

3. Argumentation Process:
   – Whether the development of the polished paragraph follows the required "tree structure", problem, viewpoint, reasons, and evidence.
   – Regardless of the complexity of the viewpoint, whether the viewpoint information is effectively conveyed to the reader?

4. Literature and Examples:
   – Avoid irresponsible citations, incorrect citations, counterfactual references, or irrelevant citations.

5. Cannot Discern Quality Difference

# I  PROMPTS FOR ALL EXPERIMENT SETTINGS

| Setting | Prompt |
|---|---|
| **Chinese prompt** | 你是一位专业的写作老师，你正在教授一位同学论述性写作，同学提交了他的论文草稿，请你根据你制定的以下标准，对论文草稿中的一段话进行点评。
[评价标准开始]
选题要求
* 选题基于明确的研究空白；
* 需具备学科深度、新颖性和研究价值；
* 研究对象和视角应聚焦明确。
文献使用
* 文献检索应充分且符合CRAAP原则（时效性、相关性、权威性、准确性、无利益冲突）；
* 根据具体研究问题平衡使用前沿文献和经典文献；
* 与文献进行充分对话，深入理解选题及方法，合理运用文献进行观点论证。
观点论证
* 观点明确，论据充分；
* 论证逻辑严密，结合演绎与归纳，呈现清晰的树形结构和塔式积木式的证据链。
结构组织
* 内容有清晰的主线，条理分明；
* 概念、框架应前后一致，文内合理呼应；
* 通过标题、段首词句衔接，实现流畅过渡；
* 论点前置，吸引读者，回答"为什么写、写了什么、怎么写"；
* 结尾自然，无不必要的评论或总结。
规范与语言
* 遵守学术规范，论证应为原创，合理引用而非照搬；
* 排版整洁，符合模板要求，引用符合标准格式；
* 语言准确、简洁、理性，避免使用个人化表达。
[评价标准结束]

以下是学生的作文，你需要先阅读并理解其内容。

[学生作文开始]
{essay}
[学生作文结束]

下面是需要你评价的局部段落，请你在评价的时候定位其在文章中的位置。

[待评价段落开始]
{paragraph}
[待评价段落结束]

现在请开始你的评价。请注意，评价标准中列出了5点要求，但是你只需要根据待评价文段中最明显的特征，在选题要求、文献使用、观点论证、结构组织、规范与语言中选取一个维度进行评价即可。你的评价旨在帮助学生提升待评价段落的质量，如果待评价段落中存在问题，请将其指出，并且提供改进建议。请直接回复你的评价，不要套话。你的评价不要超过100字。 |
| **English translation** | You are a professional writing instructor teaching a student argumentative writing. The student has submitted a draft of their essay. Based on the following criteria, please provide feedback on a specific paragraph from the draft.

[Evaluation Criteria Begins]
Topic Selection Requirements
- The topic should be based on a clear research gap.
- It should have academic depth, novelty, and research value.
- The research object and perspective should be focused and specific.
Use of Literature
- The literature search should be thorough and meet the CRAAP principles (Currency, Relevance, Authority, Accuracy, and Purpose).
- Use a balanced mix of cutting-edge and classic literature, depending on the research question.
- Engage deeply with the literature to understand the topic and methodology, and use it appropriately to support arguments.
Argumentation
- The argument should be clear, with sufficient evidence.
- The logic should be rigorous, combining deduction and induction, presenting a clear tree structure and a block-by-block evidence chain.
- Structure and Organization
The content should follow a clear main line, well-structured.
- Concepts and frameworks should be consistent and logically referenced throughout the essay.
- Smooth transitions should be achieved through appropriate use of headings and introductory phrases.
- The thesis should be upfront, engaging the reader, answering "why write, what is written, how it is written."
- The conclusion should be natural, without unnecessary commentary or summary.
Academic Norms and Language
- Follow academic standards; arguments should be original, with proper citation instead of paraphrasing or copying.
- The formatting should be neat, adhering to template requirements, and citations should follow the correct format.
- The language should be accurate, concise, and objective, avoiding personal expressions.
[Evaluation Criteria Ends]

Here is the student's essay; please read and understand its content first.

[Student Essay Begins]
{essay}
[Student Essay Ends]

Below is the specific paragraph to be evaluated. When providing feedback, please identify its position in the essay.

[Paragraph to be Evaluated Begins]
{paragraph}
[Paragraph to be Evaluated Ends]

Now, please provide your evaluation. Note that although five aspects are listed in the evaluation criteria, you only need to evaluate one dimension based on the most prominent feature in the paragraph. Your feedback should help the student improve the quality of the paragraph. If there are issues, please point them out and offer suggestions for improvement. Please respond with your feedback directly without using formalities, and your evaluation should not exceed 100 words. |

Table 10: Prompt for critiquing essays directly based on essays and paragraphs with **zero-shot base LLMs** in Chinese.

| Setting | Prompt |
|---|---|
| **Chinese prompt** | 你是一位专业的写作老师，你正在教授一位同学论述性写作，同学提交了他的论文草稿，请你根据你制定的以下标准，对论文草稿中的一段话进行点评。
[评价标准开始]
选题要求
* 选题基于明确的研究空白；
* 需具备学理深度、新颖性和研究价值；
* 研究对象和视角应聚焦明确。
文献使用
* 文献检索应充分且符合CRAAP原则（时效性、相关性、权威性、准确性、无利益冲突）；
* 根据具体研究问题平衡使用前沿文献和经典文献；
* 与文献进行充分对话，深入理解选题及方法，合理运用文献进行观点论证。
观点论证
* 观点明确，论据充分；
* 论证逻辑严密，结合演绎与归纳，呈现清晰的树形结构和塔式积木式的证据链。
结构组织
* 内容有清晰的主线，条理分明；
* 概念、框架应前后一致，文内合理呼应；
* 通过标题、段首词句衔接，实现流畅过渡；
* 论点前置，吸引读者，回答"为什么写、写了什么、怎么写"；
* 结尾自然，无不必要的评论或总结。 规范与语言
* 遵守学术规范，论证应为原创，合理引用而非照搬；
* 排版整洁，符合模板要求，引用符合标准格式；
* 语言准确、简洁、理性，避免使用个人化表达。
[评价标准结束]
以下是是学生的作文，你需要先阅读并理解其内容。

[学生作文开始]
{essay}
[学生作文结束]
为了更好地理解这篇文章的内容，你带着几个主要问题阅读文章，并且得到了对文章的总体认识。下面是你的问题和相应回答：
[你的笔记开始]
{qa_notes}
[你的笔记结束]
下面是需要你评价的局部段落，请你在评价的时候定位其在文章中的位置。
[待评价段落开始]
{paragraph}
[待评价段落结束]

现在请开始你的评价。请注意，评价标准中列出了5点要求，但是你只需要根据待评价文段中最明显的特征，在选题要求、文献使用、观点论证、结构组织、规范与语言中选取一个维度进行评价即可。在你的评价过程中，请你结合你的笔记，把握文章的整体框架、思路、逻辑。你的评价旨在帮助学生提升待评价段落的质量，如果待评价段落中存在问题，请将其指出，并且提供改进建议。请直接回复你的评价，不要套话。你的评价不要超过100字。 |
| **English translation** | You are a professional writing instructor teaching a student argumentative writing. The student has submitted a draft of their essay. Based on the following criteria, please provide feedback on a specific paragraph from the draft.

[Evaluation Criteria Begins]
Topic Selection Requirements
- The topic should be based on a clear research gap.
- It should have academic depth, novelty, and research value.
- The research object and perspective should be focused and specific.
Use of Literature
- The literature search should be thorough and meet the CRAAP principles (Currency, Relevance, Authority, Accuracy, and Purpose).
- Use a balanced mix of cutting-edge and classic literature, depending on the research question.
- Engage deeply with the literature to understand the topic and methodology, and use it appropriately to support arguments.
Argumentation
- The argument should be clear, with sufficient evidence.
- The logic should be rigorous, combining deduction and induction, presenting a clear tree structure and a block-by-block evidence chain.
- Structure and Organization
The content should follow a clear main line, well-structured.
- Concepts and frameworks should be consistent and logically referenced throughout the essay.
- Smooth transitions should be achieved through appropriate use of headings and introductory phrases.
- The thesis should be upfront, engaging the reader, answering "why write, what is written, how it is written."
- The conclusion should be natural, without unnecessary commentary or summary.
Academic Norms and Language
- Follow academic standards; arguments should be original, with proper citation instead of paraphrasing or copying.
- The formatting should be neat, adhering to template requirements, and citations should follow the correct format.
- The language should be accurate, concise, and objective, avoiding personal expressions.
[Evaluation Criteria Ends]
Here is the student's essay; please read and understand its content first.
[Student Essay Begins]
{essay}
[Student Essay Ends]
To better understand the content of this article, you read it with several key questions in mind, gaining an overall insight into the work. Below are your questions and their corresponding answers:
[Your notes begin]
{qa_notes}
[Your notes end]
Below is the specific paragraph to be evaluated. When providing feedback, please identify its position in the essay.
[Paragraph to be Evaluated Begins]
{paragraph}
[Paragraph to be Evaluated Ends]
Now, please provide your evaluation. Note that although five aspects are listed in the evaluation criteria, you only need to evaluate one dimension based on the most prominent feature in the paragraph. In your evaluation, please integrate your notes to grasp the overall framework, thought process, and logic of the article. Your feedback should help the student improve the quality of the paragraph. If there are issues, please point them out and offer suggestions for improvement. Please respond with your feedback directly without using formalities, and your evaluation should not exceed 100 word. |

Table 11: Prompt for critiquing essays using **RedHat**. It reserve a field 'qa_notes' for the question-anwering results.

| Setting | Prompt |
|---|---|
| **Chinese prompt** | 请扮演一位专业的论文评审专家，在读懂论文的基础上判断两条评语的质量。请先阅读以下的长文。

[文章开始]
essay
[文章结束]

下面是一对评语与评语对应的段落，请你判断哪一条评语质量更好。评语的质量好坏主要体现在：
1. 评语是否理解了段落的内容，特别是在作者的写作意图基础上展开的；
2. 评语是否足够深入，特别是对改进段落质量有帮助
3. 评语是否避免了幻觉，例如事实错误，逻辑错误，过分解读，不理解文本本身等；

[段落开始]
{paragraph}
[段落结束]

[评语1开始]
{comment1}
[评语1结束]

[评语2开始]
{comment2}
[评语2结束]

你需要给出四种判断之一：1更好；2更好；1和2一样好；1和2一样差。请以两对中括号包括你的回答，例如"[[1更好]]"，或者"[[2更好]]"等。请直接给出你的判断。 |
| **English translation** | Please act as a professional paper reviewer and assess the quality of two comments based on your understanding of the paper. First, read the following text.

[Article begins]
essay
[Article ends]

Below is a pair of comments along with the corresponding paragraph. Please determine which comment has better quality. The quality of the comments is primarily evaluated based on:
1. Whether the comment accurately understands the content of the paragraph, especially in relation to the author's intent;
2. Whether the comment is sufficiently in-depth, particularly in its usefulness for improving the quality of the paragraph;
3. Whether the comment avoids misconceptions, such as factual errors, logical fallacies, over-interpretation, or misinterpretation of the text itself.

[Paragraph begins]
{paragraph}
[Paragraph ends]

[Comment 1 begins]
{comment1}
[Comment 1 ends]

[Comment 2 begins]
{comment2}
[Comment 2 ends]

You need to provide one of four judgments: 1 is better; 2 is better; 1 and 2 are equally good; 1 and 2 are equally poor. Please enclose your answer in double brackets, such as "[[1 is better]]" or "[[2 is better]]". Please provide your judgment directly. |

Table 12: Prompt for GPT-4o-mini-0718 to **compare the critique quality** between the polished texts with different critiques.

| Setting | Prompt |
|---------|--------|
| **Chinese prompt** | 请扮演一位专业的论文润色专家，在读懂论文的基础上，结合你的阅读笔记，以及一个评阅意见，对一个段落进行润色、优化。

[文章开始]
{essay}
[文章结束]

[阅读笔记开始]
{notes}
[阅读笔记结束]

[段落开始]
{paragraph}
[段落结束]

[评语开始]
{critique}
[评语结束]

依据评语，请直接写出你改进后的段落，不需要其他说明。 |
| **English translation** | Please act as a professional paper editing expert. After fully understanding the paper, and based on your reading notes as well as a critique, revise and optimize a given paragraph.

[Start of Essay]
{essay}
[End of Essay]

[Start of Reading Notes]
{notes}
[End of Reading Notes]

[Start of Paragraph]
{paragraph}
[End of Paragraph]

[Start of Critique]
{critique}
[End of Critique]

Based on the critique, please directly write the improved paragraph without any further explanation. |

Table 13: Prompts for instructing GPT-4o-0806 to **polish the original text** based on the critique.

| Setting | Prompt |
|---|---|
| **Chinese prompt** | 请扮演一位专业的论文评审专家，读懂论文的基础上比较一段话不同润色结果的的质量。请先阅读以下的长文。

[文章开始]
{essay}
[文章结束]

下面是一对润色结果与原文段落，请你判断哪一条润色结果质量更好。润色结果的质量好坏主要体现在：
1. 放入原文中的位置是否通顺、合理，在文章片段上表意连贯、思路清晰；
2. 本身没有明显可见的事实错误、论述不当；
3. 使得文章结构更加完整，不明显偏离原文主线。

[原文开始]
{paragraph}
[原文结束]

[润色结果1开始]
{polish1}
[润色结果1结束]

[润色结果2开始]
{polish2}
[润色结果2结束]

你需要给出四种判断之一：1更好；2更好；1和2一样好；1和2一样差。请以两对中括号包括你的回答，例如"[[1更好]]"，或者"[[2更好]]"等。请直接给出你的判断。 |
| **English translation** | Please act as a professional paper reviewer and assess the quality of different revisions of a paragraph based on your understanding of the paper. First, read the following text.

[Article begins]
{essay}
[Article ends]

Below is a pair of revisions compared to the original paragraph. Please determine which revision has better quality. The quality of the revisions is primarily evaluated based on:
1. Whether the placement of the revisions within the original text is coherent and reasonable, maintaining a clear flow of ideas;
2. The absence of obvious factual errors or inappropriate arguments;
3. The enhancement of the overall structure of the paper without significantly deviating from the original main line.

[Paragraph begins]
{paragraph}
[Paragraph ends]

[Revision result 1 begins]
{polish1}
[Revision result 1 ends]

[Revision result 2 begins]
{polish2}
[Revision result 2 ends]

You need to provide one of four judgments: 1 is better; 2 is better; 1 and 2 are equally good; 1 and 2 are equally poor. Please enclose your answer in double brackets, such as "[[1 is better]]" or "[[2 is better]]". Please provide your judgment directly. |

Table 14: Prompt for GPT-4o-0806 to **compare the quality between the polished texts** with different critiques.

