# OpenReview forum: "RedHat: Towards Reducing Hallucination in Essay Critiques with Large Language Models"
_ICLR.cc/2025/Conference — Submitted to ICLR 2025_

### Official Review · Reviewer_rgUd · 2024-10-29

**Soundness:** 2
**Presentation:** 3
**Contribution:** 2
**Rating:** 8
**Confidence:** 4

**Summary:**

The background proposed in this paper is to use LLM to evaluate and criticize essay writing. It points out that the efficiency of using LLM is much higher than that of humans. However, LLM has some problems in performing this task, such as (1) Providing suggestions that do not match or are inappropriate to the essay content, and (2) Proposing logical errors that do not exist in the essay. These hallucination seriously affect the usability of LLMs in generating essay reviews.  This paper has two main contributions, a dataset and a new method to mitigate hallucinations. The results of automatic and manual evaluation demonstrate the effectiveness of the work proposed by the authors.

**Strengths:**

1. The authors list definitions of various types of hallucinations in Table 1, which is helpful for clearly defined tasks.
2. The evaluation setting is solid which including both auto and human evaluation.

**Weaknesses:**

1. From Figure 4, it seems that the evaluation results of GPT are not reliable. This is because people trust the results of artificial experts more. Although the results of untrained human annotators are also random.
2. Why consider ROUGE and BLUE as metrics, as they only consider word overlap. Maybe try BERTScore, BLEURT.
3.The writing structure seems a bit confusing. Chapters 2, 3, and 4 each contain a literature review, plus the work of this paper.
4. It is very good that the authors present a new dataset. But perhaps more annotation process and quality assessment should be provided?

**Questions:**

Q1: From the perspective of the problem defined in Figure 1, if LLM fails to focus on the desired argument or is too focused on the conclusion, is it fit to be called a hallucination?
Q2: I am not sure whether such an open Q-A result is stable. Will it change with each run of model?

---

> ### Author Response · Authors · 2024-11-25
> **Responses to Reviewer rhUd (1/2)**
>
> We greatly appreciate your reviews and valuable suggestions. We have carefully addressed each of your concerns in detail.
>
> > W1-1: reliability of GPT and human annotations
>
> Thanks for your comments. We have added Qwen-2-7b-Instruct on EssayC along with ChatGPT-4o on a subset of artificial intelligence conference papers. Please refer to General Response 1 for details. In the Qwen experiment, humans are more inclined to believe that RedHat has improved the critiques' quality (around 7-10% more preferred on RedHat).
>
> Sorry for not being clear enough with the previous performance presentation. We sorted out the experiment data, including Redhat is good, baseline-LLM is good, both are good and both are bad. We found that the performance of RedHat is more preferred by humans compared to baseline-LLM in both Qwen and GLM. Therefore, we believe that Redhat made improvements when judged by both humans and GPT-4. We have redrawn Figure 4, split the contents of the tie, and added Qwen’s experiment results.
>
> |                   | RedHat better | Baseline better | Both are good | Both are bad |
> | :---------------: | :-----------: | :-------------: | :-----------: | :----------: |
> |   Qwen2-redhat    |     33.26     |      26.04      |     7.38      |    33.32     |
> | Qwen2-redhat-weak |     36.30     |      25.94      |     7.18      |    30.57     |
> |    GLM4-redhat    |     32.49     |      25.38      |     8.38      |    33.76     |
> | GLM4-redhat-weak  |     32.49     |      28.43      |     7.61      |    31.22     |
>
> > W2-1: why word-level overlap metrics & consider BLEURT and BERTScore
>
> Thank you for your valuable insight. We have chosen ROUGE and BLEU to verify that the embedded questions do not show leakage of desirable contents in explicit n-gram forms. Our results also verify that the questions are not similar to the final RedHat-generated critiques at the word level.
>
> We have added BLEURT and BERTScore as similarity scores and updated them in the revised version. We quote the results in the tables below.
>
> - The similarity between the critique with the questions (column 1) and the answers (column 2) are presented. Comparing the baseline (Row 1) with RedHat (Rows 2-3), there is no significant change in the similarity of the question, but the critique generated by RedHat is clearly more similar to the answer. This confirms that the gain in evaluation is benefited from the answer inferred by the LLMs themselves, rather than by the criteria-derived question. We have corrected this confusion in our revised version.
>
> |       BLEURT       | Questions | $\Delta$ | Answers | $\Delta$ |
> | :----------------: | :-------: | :------: | :-----: | :------: |
> | Qwen-2-7b-Instruct |   27.45   |    0     |  21.39  |    0     |
> |      +redhat       |   25.24   |  -2.21   |  31.00  |  +9.61   |
> |    +redhat-weak    |   26.24   |  -1.21   |  29.41  |  +8.02   |
>
> |     BertScore      | Questions | $\Delta$ | Answers | $\Delta$ |
> | :----------------: | :-------: | :------: | :-----: | :------: |
> | Qwen-2-7b-Instruct |   71.09   |    0     |  74.50  |    0     |
> |      +redhat       |   72.39   |  +1.30   |  76.16  |  +1.66   |
> |    +redhat-weak    |   72.50   |  +1.41   |  76.41  |  +1.91   |
>
> > W2-2: paper writing that Chapters 2, 3, and 4 each contain a literature review with the work of this paper.
>
> We sincerely appreciate your suggestions. We have moved the Related Work section to the appendix. Specifically, the second section formally defines the essay critiquing task, the third section illustrates hallucinations in our task, and the fourth section introduces our method to mitigate hallucinations. At the beginning of each paragraph, we have briefly described its content, and we have also added a description of the paper's structure in the introduction section to improve readability.
>
> > W3: annotation process and quality assessment of the data resource
>
> Thanks for your suggestion. Please refer to General Response 2 for the  detailed description of EssayC. We have introduced the annotation process and the quality assessment in the *Filter* and *Annotation and quality assessment* part.

---

> ### Author Response · Authors · 2024-11-25
> **Responses to Reviewer rhUd (2/2)**
>
> > Q1: hallucination definition. Whether an LLM fails to focus on the desired argument or is too focused on the conclusion, is it fit to be called a hallucination?
>
> As we have discussed in Section 3. The core of essay critiques can be divided into hallucinating non-existing writing problems and overlooking the essay's logic and structure with overemphasizing details, or vague comments with no in-depth advice.
>
> We discuss the above mentioned question in two cases:
>
> - Not hallucination: if the desired argument is important in the logic frame of the essay, such focus is needed and the supporting evidence and logical structure should be inspected before LLMs come to a final critique.
> - Misunderstand the author’s perspective (Type I hallucination): if one perspective is first introduced in the introduction (opening) of the essay, the author is not expected to expand it with arguments - in other words, LLM fails to understand the author's intention or perspective if it is focusing on the arguments.
>
> We have also made a comparison with our definition of hallucination with previous works in summarization, knowledge QA and reasoning.
>
> > Q2: Stability of the questions with each run of model
>
> Thank you for your reminder. We repeated using GPT-4o to generate a question list from the evaluation criteria. We conducted 5 trials, asking GPT-4o to generate 20 question-answers each time. We compared the latter 4 trials's recall rates to the first trial as consistency listed below. Most of the questions are the same and it validates stability.
>
> | Trial       | 1    | 2    | 3    | 4    | 5    |
> | ----------- | ---- | ---- | ---- | ---- | ---- |
> | Consistency | 100% | 95%  | 95%  | 100% | 95%  |

---

> ### Author Response · Authors · 2024-11-29
> **We look forward to your feedback**
>
> Dear Reviewer rgUd,
>
> The authors are grateful for your insightful comments. Since there are three days left until the discussion period ends, we are unsure whether the response above has fully addressed your concerns. If there are any remaining issues, please feel free to contact us. We sincerely look forward to your feedback.
>
> Best regards,
>
> The authors

---

> ### Comment · Reviewer_rgUd · 2024-12-03
> **Thank you for response**
>
> Thank you for author's response! I think what the authors are missing is not how humans assess the performance of these models, but rather whether the work of the humans is reliable. You should calculate metrics like Cohen's kappa to evaluate the consistency of the human annotators' work.
>
> That said, I believe the authors have addressed most of my concerns, and I hope they can consider incorporating this metric in the future. I have decided to raising my score.

---

> ### Author Response · Authors · 2024-12-03
> **Further discussion on Inter-Annotator Agreement**
>
> We deeply appreciate your recommendation of our work and your advise to compute the Cohen's Kappa as indicator of inter-annotator agreement (IAA for short).
>
> **We have calculated the IAA for each number from the main experiment annotation results and have listed in the following table**. We calculate the initial annotation agreements between two annotators before the introduction of the third annotator to recheck the disagreed results. The results are listed in the following table. Most of the agreements fall in intervals [0.4, 0.6) and [0.6, 0.8), i.e. between moderate/substantial agreements. The overall score's IAA is generally lower than dimensional ones, since it has up to 6 categories. The dimensional scores are mainly substantially significant. Results indicate that most annotators agree with each other in the first round of double check and there are certain levels of subjectivity during the process.
>
> |              | Overall | Hallu | Ambig | Info |
> | ------------ | ------- | ----- | ----- | ---- |
> | Human        | 0.64    | 0.75  | 0.81  | 0.71 |
> | GLM          | 0.52    | 0.64  | 0.69  | 0.63 |
> |   +RedHat      | 0.57    | 0.67  | 0.72  | 0.65 |
> |   +RedHat-weak | 0.54    | 0.66  | 0.71  | 0.65 |
> |   +SFT         | 0.51    | 0.63  | 0.67  | 0.58 |
> |   +SFT-RedHat  | 0.52    | 0.64  | 0.70  | 0.59 |
> |   +PT          | 0.58    | 0.66  | 0.71  | 0.63 |
> | Qwen         | 0.54    | 0.58  | 0.69  | 0.53 |
> |   +5-shot     | 0.57    | 0.62  | 0.61  | 0.56 |
> |   +RedHat      | 0.55    | 0.63  | 0.74  | 0.67 |
> |   +RedHat-weak | 0.58    | 0.63  | 0.72  | 0.67 |
> |   +SFT         | 0.55    | 0.56  | 0.65  | 0.65 |
> |   +SFT-RedHat  | 0.49    | 0.58  | 0.69  | 0.66 |
> |   +PT          | 0.48    | 0.62  | 0.63  | 0.67 |
> | ChatGPT      | 0.57    | 0.65  | 0.76  | 0.66 |
> |   +RedHat   | 0.53    | 0.68  | 0.70  | 0.62 |

---

### Official Review · Reviewer_ssM2 · 2024-10-29

**Soundness:** 3
**Presentation:** 3
**Contribution:** 2
**Rating:** 5
**Confidence:** 3

**Summary:**

This paper addresses the essay critique generation task using large language models (LLMs) and proposes a technique to reduce hallucination in this task. The authors propose RedHat, a technique to reduce hallucination by incorporating document-level question-answering into the critique generation process. The authors compare the proposed method with others, including post-pretraining (PT) and supervised fine-tuning (SFT), and show that their proposed method does reduce hallucination.

**Strengths:**

* The proposed method, RedHat, appears to be grounded in solid assessment practices, with critiques based on concrete practices in answering questions.
* They are working with a non-English (Chinese) dataset and task.
* They provide a clear, position-based analysis highlighting where improvements occurred.

**Weaknesses:**

* The proposed method, RedHat, simply adds document-level questions and answers, which may come across as an incremental, somewhat trivial improvement.
* It’s unclear whether the issue the authors address is truly "hallucination" or simply improving task performance in critique generation. Are there types of essay critique generation errors that are not hallucination? A clear definition would be valuable here.
* The paper is quite vague; while the tasks and datasets used are in Chinese, this isn’t clearly stated. Working on non-English tasks and datasets is generally beneficial for diversity in the English-dominated LLM landscape. However, the paper lacks clarity on this and, more importantly, offers very limited insights on whether the findings apply to other datasets, languages, or base LLMs (e.g., GLM). For instance, it would be useful to know if similar techniques could improve essay critique generation in English using models like ChatGPT.

**Questions:**

* There's very little information about EssayC. Including an overview of the dataset (e.g., types and size of data) would be helpful. Additionally, will you open-source the dataset?
* There’s limited information about the baseline methods (e.g., SFT and PT) in the main body of the paper, particularly regarding the training data used. It was challenging to navigate between the main paper and the appendix to find these details.
* You might consider a few-shot approach, where a few examples of essays and their critiques are provided in the prompt. This approach is often effective for tasks like essay scoring for non-native speakers.
* In Section 5.1, the metric "Detailedness" appears to be lower-is-better, while in the objective function (2) it is being maximized. Perhaps renaming it to something like "ambiguity" would clarify its intent.

---

> ### Author Response · Authors · 2024-11-25
> **Response to Reviewer ssM2 (1/3)**
>
> Thank you for your thorough review and valuable feedback! We have worked  to address your concerns one by one.
>
> > W2: unclear whether the issue the authors address is truly "hallucination"  & types of errors not regarded as hallucination
>
> In our work, LLM-generated essay critiques mainly face two types of problems. Type I: Hallucinate writing problems that do not exist in the essay. Type II: Overlook the essay's logic and structure by overemphasizing on details, or just vague comments with no in-depth advice. We find the two types are consistent with hallucinations discussed in previous literatures. Specifically speaking,
>
> - For type I, [1] mentioned "ignore the source material altogether" as *Extrinsic Hallucination* in summarization task. [2] defined "generation of nonfactual, untruthful information, especially ignoring the context"  in ChatGPT's reasoning and multimodal tasks.
> - For type II, [1] mentioned "misrepresent information from the doc" as *Instrinsic Hallucination* in summarization task, [3] concluded "Generation of plausible looking statements that are factually incorrect" in conversation task.
>
> Considering the features of our type I and II hallucinations, our definition agrees with hallucination definition in summarization, conversation, QA, reasoning tasks. Based on the discussion above, we lend the wording **hallucination** to describe our phenomenon. We present the discussed literature in the following Table 5 for further interest.
>
> | Work                 | Citation | Task                              | Definition                                                   |
> | -------------------- | -------- | --------------------------------- | ------------------------------------------------------------ |
> | Maynez et al., 2020  | 1180     | Summarization (RNN)               | (On Page 3) Intrinsic Hallu: misrepresent info of from the docExtrinsic Hallu: Model ignore the source material altogetherFactual Hallu: Contain info not found in doc ( composed of instrinsic and extrinsic hallu) |
> | Shuster et al., 2021 | 592      | Knowledge Conversation (BART, T5) | (On Page 1) Generation of plausible looking statements that are factually incorrect |
> | Bang et al., 2023    | 1301     | QA, reasoning (ChatGPT)           | (On Page 3) Generation of nonfactual, untruthful information, mainly extrinsic in Work 1 |
>
> In our work, errors that do not relate to the factuality of the original essays are not regarded as hallucinations. Here are two examples: (1) segmentation, that the critique is not presented in complete sentences; (2) tone or roleplay, whether the critique is expressed in the tone of an instructor or an assistant. They are beyond the scope of our work and also do not dominate in practice, and we did not include them in our category.
>
> [1] Joshua Maynez et al. On faithfulness and factuality in abstractive summarization. ACL. 2020
>
> [2] Kurt Shuster et al. Retrieval augmentation reduces hallucination in conversation. EMNLP. 2021
>
> [3] Yejin Bang et al. A multitask, multilingual, multimodal evaluation of chatgpt on reasoning, hallucination, and interactivity. IJCNLP. 2023

---

> ### Author Response · Authors · 2024-11-25
> **Response to Reviewer ssM2 (2/3)**
>
> > W1: The proposed method, RedHat, simply adds document-level questions and answers, which may come across as an incremental, somewhat trivial improvement.
>
> We appreciate the reviewer's comments and valuable suggestions. As we discussed in Section 3, LLM-generated essay critiques mainly face two types of problems.
>
> - Type I:  Hallucinating writing problems that do not exist in the essay.
> - Type II: Overlook the essay's logic and structure by overemphasizing on details, or just vague comments with no in-depth advice.
>
> We identify them as hallucinations as [1], [2], [3] did in summarization, knowledge conversation, QA and reasoning tasks. Yet there is a gap in research addressing essay critique generation, especially prompt engineering, like adjusting the order of presenting inputs, few-shot or in-context learning, SFT method cannot reduce hallucination, indicating critique hallucination is not caused by task mis-alignment. We discover that it is closely related to the un-groundness of LLM to the target essay.
>
> We have curated RedHat, aiming at improving LLM's awareness of the target essay's perspectives and logic structure. On the one hand, RedHat facilitates the assessment process into (1) proposing questions crucial for evaluating the essay; (2) answering questions based on the evidence from the essay, and (3) feeding the above digest to the evaluation instruction. RedHat harnessed LLM's ability to improve its awareness of the details and logical structure of the essay.  On the contrary, SFT methods on the polished human critiques result in heavy hallucination (10% drop in hallucination in the main experiment), and few-shot methods have less effectiveness in results compared to RedHat.
>
> To further prove our solution's effectiveness, we added experiments with Qwen-2-7b-Instruct with EssayC, and ChatGPT-4o with Computer Science Conference papers (mainly ICLR, Neurips, ACL). Results are shown in General Response 1. RedHat shows its effectiveness in both experiments, both in point-wise scoring and preference comparison. This also indicates the generalizability of our methods to more models and writing genres.
>
> [1] Joshua Maynez et al. On faithfulness and factuality in abstractive summarization. ACL. 2020
>
> [2] Kurt Shuster et al. Retrieval augmentation reduces hallucination in conversation. EMNLP. 2021
>
> [3] Yejin Bang et al. A multitask, multilingual, multimodal evaluation of ChatGPT on reasoning, hallucination, and interactivity. IJCNLP. 2023
>
>
>
> > W3 & Q1: Vague on dataset & whether the findings apply to other datasets, languages, or base LLMs
>
> We appreciate your suggestion. We have added the description of the EssayC dataset in Section 2 to echo the details in the revised paper. Please refer to General Response 2 for more information about EssayC.
>
> We have also added experiments with Qwen-2-7b-Instruct on EssayC as well as with ChatGPT-4o on artificial intelligence conference papers. The details can be found in General Response 1. Briefly speaking, results from the Qwen-2 model on EssayC and from ChatGPT-4o on the English artificial intelligence conference paper are consistent with the previous experiment. (GLM-4 on EssayC), which demonstrates that our findings can be applied to further datasets, languages and base LLMs.
>
> > Q1: Will you open-source the dataset?
>
> If accepted, we will make the dataset and manual annotation results public based on the wishes of the undergraduate essay authors and graduate teaching assistant annotators. We regard the resource having the potential to develop automatic evaluation and alignment methods for automatic essay critique generation.

---

> ### Author Response · Authors · 2024-11-25
> **Response to Reviewer ssM2 (3/3)**
>
> > Q2: baseline methods information on training data for SFT and PT
>
> We apologize for the confusion and inconvenience. We have made revisions to the manuscripts and briefly introduce the training data of supervised finetuning (SFT) and post-pretraining (PT) as follows.
>
> For SFT, we adopt 31,694 polished human paragraph-level critiques as training data, excluded from the EssayC testset split (please refer to General Response 2 for more details). The format of the data is arranged into (eval prompt, essay, and target paragraph) as input, and polished paragraph as output. The train and valid set are split based on essays to avoid potential leakage.
>
> As for post-pretraining, we follow two steps: (1) pre-training on Chinese academic papers in the field of literature, social science and humanities and (2) followed by SFT on the previous data to ensure the alignment of the critiquing task. We crawled 128,321 academic papers from the Chinese National Social Science Base. The papers mainly come from journals, such as *Exploration and  Free Views, Fiction Monthly Shanghai Literature, Beijing Literature Novella Month, Science Technology Critiques, Tanzhen Technology Review and so on*. We use OCR with doc2x API (https://v2.doc2x.noedgeai.com) and applied the follow-up data filter and typo fixing with GPT and GLMs. The above process produces 27,430 pure text papers of an average of 30,000 Chinese characters. The whole tokens surpassed 1.5 billion.
>
> > Q3: validate the few-shot approach on this task
>
> We sincerely appreciate your reminder of the few-shot methods. We conducted a few-shot (5-shots) experiment in the Qwen-2 experiments and the results can be found in General Response 1. We highlight the comparison with RedHat below. The results showed that 5-shots exhibited positive improvements in the 4 metrics, and the improvements are more evident. Yet, there is a gap between both of the two methods and human critiques.
>
> |                   | Overall % ($\uparrow$) | Hallucination % ($\downarrow$) | Ambiguity % ($\downarrow$) | Informative % ($\uparrow$) |
> | ----------------- | ---------------------- | ------------------------------ | -------------------------- | -------------------------- |
> | Qwen2             | 3.187                  | 62.53                          | 14.68                      | 11.90                      |
> | 5-shots           | 3.178                  | 61.01                          | 11.14                      | 12.66                      |
> | Qwen2-redhat      | 3.267                  | 62.03                          | 7.59                       | 15.70                      |
> | Qwen2-redhat-weak | 3.323                  | 58.73                          | 8.10                       | 18.73                      |
> | Qwen2-sft         | 2.615                  | 74.43                          | 27.59                      | -19.24                     |
> | Qwen2-PT          | 2.777                  | 71.65                          | 24.30                      | -9.11                      |
> | Human             | 3.387                  | 47.34                          | 11.65                      | 30.63                      |
>
> > Q4: Renaming "Detailedness" to "Ambiguity"
>
> We sincerely appreciate your suggestion. In Section 2 we change the objective function to minimize ambiguity, as well as the changed the notation of D(c) to A(c). Also, we rephrase the "Detailedness" into "Ambiguity" in all experiment result tables.

---

> ### Author Response · Authors · 2024-11-29
> **We look forward to your feedback**
>
> Dear Reviewer ssM2,
>
> The authors appreciate your efforts in improving the manuscript. Since there are three days left until the discussion period ends, we are unsure whether the responses above has fully addressed your concerns. If there are any remaining issues, please feel free to contact us. We are looking forward to your reply.
>
> Best regards,
>
> The authors

---

> ### Author Response · Authors · 2024-12-03
> **Additional Experiments and Clarifications**
>
> Dear Reviewer,
>
> We are deeply grateful for your devotion to reviewing our manuscript. We have added two further experiments on different base-LLMs (Qwen2 and GPT) and languages (English). The extra experimental statistical results have also indicated the effectiveness of RedHat. We also have made clarification on concept definition, training data construction, the mechanism of our methodology in reducing the hallucination in critiques and other details. Hope our updated results resolve your concerns.
>
> As this may be our final opportunity to address any remaining concerns you may have, if there are any aspects of our paper that require further clarification, please don't hesitate to inform us. We eagerly await your insights and the opportunity to engage in a constructive discussion to strengthen our submission. Thank you!
>
> Best regards,
>
> The authors

---

### Official Review · Reviewer_oeU8 · 2024-11-05

**Soundness:** 2
**Presentation:** 3
**Contribution:** 2
**Rating:** 3
**Confidence:** 4

**Summary:**

The paper introduces a benchmark and corresponding experiments for the task of automatically providing useful free-form feedback for human-written essays. The main objective is to reduce to what the paper refers to as 'hallucinations' - i.e. in this case irrelevant feedback.
To achieve this, the paper introduces a resource called EssayC - essays written in Chinese by students, which are enriched by LLM-generated critiques. The main contribution of the paper seems to be the use of an automatically derived questionnaire aimed at evaluating different aspects pertaining to the quality of the essay, which - together with automatically derived answers - is used to enrich the prompt of LLMs when generating the essay feedback, an approach which the papers calls "RedHat".

**Strengths:**

Overall this work is intriguing and makes good use of prompt engineering in a clever way. The results seems to indicate some level of success.

**Weaknesses:**

I have a few concerns with the paper:

- Firstly, the construction of the evaluation criteria seems rather ad-hoc, GPT4 is prompted for questions, which are then voted upon by human experts in form of GTAs. The results are in no way linked to or rooted in existing rich literature on evaluating essays. I think this is a missed opportuninty to investigate, how well GPT4's question lists align with existing criteria proposed in the literature. Similarly, the generated critiques are not compared to the actual human-written critiques which existed for the collected essays, in fact they were deliberatly removed and disregarded. This again presents a missed opportuninty to further understand the quality of the approach.

- Secondly, for all the prompt engineering, the results are not very impressive. This is down to a few points: Firstly, only a single LLM, chatGLM-9B is evaluated. Secondly, the overall scores in Table 2, pairwise comparisons in Figure 4 and Table 4 show only a small improvement over the chatGLM baseline. In fact given the rather small dataset size, I'm not convinced they are statistically significant. Thirdly, i am not at all convinced by section 5.3 - the Table and the corresponding explaining paragraph (l 428 ff) seem at odds with each other: The paragraph claims that QA-enhanced RedHat methodology increases the overlap (which in itself doesn't mean that the questions are useful), but the table shows the opposite. Indeed the baseline without QA seems to have a higher overlap than the RedHat method.

- Lastly, I am not convinced by the gravity and impact of the contribution - the task of essay feedback generation is rather niche and very application-focussed, only one language is studied and the presented approach amounts to prompt engineering. Generalising the findings would be difficult, as the human studies would need to be reproduced.

Minor concerns include the wording - for example, the decision to use the word "hallucinations" is not very spot-on, "hallucinations" refer to content that is made up in summaries, i don't think what the paper describes as "hallucinations" meets this broad definition. Furthermore, the vocabulary used to outline the contributions is too strong given the actual results - for example l 103 claims to "prove" generalisability, yet I have not found a formal proof. Similarly, l 107 states that human annotators prefer automatically refined essays based on RedHat's feedback, while the majority of the annotators is indifferent to the changes (60% vote tie in Table 4).


Overall, given the concerns outlined below, I would gravitate towards rejection.

My recommendations to improve the paper would be to (a) root the approach more firmly in existing literature e.g. by comparing the question lists with existing literature; (b) increase the scope, by e.g. comparing more models or comparing model results to human-written critiques. The collection of human ratings would also enable the creation of preference data, which could be used to more efficiently align the models (e.g. by RLHF methods) rather than just using SFT. It should also enable to learn a essay quality ranking method, e.g. by regressing the overall/hallu/detail/info scores in table 2. This should greatly increase the impact of the work, as this learned metrics could be used to predict the quality of essays _without human annotations_, thus enabling to generalise the method and reproduce the findings without reproducing the human annotations.

**Questions:**

Please try to address my points raised in weaknesses, specifically (a) lack of rooting in literature and comparison between human-written and generated feedback, (b) the discrepancies between claims and empirical evidence of efficacy of your results and (c) whether you could reproduce the results using more LLMs.

---

> ### Author Response · Authors · 2024-11-25
> **Responses to Reviewer oeU8 (1/4)**
>
> We sincerely appreciate the reviewer's devotion to reviewing our manuscript and the detailed recommendations for improvement. We address the reviewer's concerns as follows and hope the response can address your concerns.
>
> > W1: regarding existing literature on automatic essay evaluation, and human written critiques.
>
> Thanks for your comments. The core contribution of our work focuses on fine-grained essay critique generation, revealing LLM-generated critique credibility problems such as un-groundness on target essay, fantasization of non-existing problems, especially suggesting adding excessive local details, showing that LLM overlooked the main arguments and logical flow. We summarized them as hallucinations. On the basis of un-groundness and overlooking essay contents in LLM's performance, we propose RedHat to reduce the hallucination of LLM-generated critiques. Specifically:
>
> - Although LLMs are widely used for evaluation, they still face challenges in the credibility of critiques for essay reviews. We have surveyed existing work in automated essay evaluation, summarized in Table 1. Current studies either focus on scoring essays (a regression task, Row 3) or lack detailed critiques (mainly summarization of the writing, Row 4-5). Moreover, the inaccessibility of fine-grained essay critique benchmarks makes it difficult to address the reliability issue.
>
> |                    | Target                | Writing Len | Generation Format | Granuality | Generation Len |
> | ------------------ | --------------------- | ----------- | ----------------- | ---------- | -------------- |
> | RedHat: (Our Work) | Argumentative Writing | 4K-6K       | Critique          | Paragraph  | ~100           |
> | Tang et al, 2024   | ASAP-AES set #7       | 150-550     | Scores            | Whole      | Integer        |
> | Liu & Shah 2023    | AI Conference Paper   | >30K        | Reviews           | Whole      | Unlimited      |
>
> |                    | Studied wriitng num                | Open Sourced | Auxiliary Questions in prompts | Essay Criteria Dimensions                                 | Critique Quality assessment                      |
> | ------------------ | ---------------------------------- | ------------ | ------------------------------ | --------------------------------------------------------- | ------------------------------------------------ |
> | RedHat: (Our Work) | 14 trained graduate TAs, 36 essays | Yes          | Predifined by GPT              | Topics, arguments, literature, structure, expression      | Hallucination, Abiguity, Informativenss, Ranking |
> | Tang et al, 2024   | 1730 essays                        | Yes          | No                             | Ideas, Organization, Style, Conventions                   | Quadratic Weighted Kappa                         |
> | Liu & Shah 2023    | 13 short papers                    | No           | Predifined by Review guidance  | Problem definition, technical advantages and shortcomings | Case study                                       |
>
> - RedHat aims to guide self-QA based on evaluation criteria for essay writing, encouraging LLM self-reflection in the generation process to address the neglect of target essay contexts and details, thus avoiding hallucination and enhancing credibility. In our survey, our work distinguishes itself from previous studies by defining a paragraph-level granularity, along with the evaluation standard on credibility.
> - When designing metrics, we primarily considered **Hallucination** for factuality, **Ambiguity** for detailedness, **Informativeness** for usefulness, and **Overall** for overall credibility. In our experiment comparisons, RedHat outperformed the baselines up to 2.03%, 5.26%, 6.6%, and 0.137 (Hallu, Ambiguity, Info, Overall), and surpassed SFT by 15.75%, 6.86%, 44.92%, and 0.824, demonstrating the effectiveness of RedHat across these aspects of credibility. Additionally, manual verification showed that the QA accuracy rates were 95.5% for GPT-4 and 85.6% for GLM-4, confirming the effectiveness of the QA mechanism in the process.
> - We would like to clarify that human feedback has been fully utilized in our work. The training objective of SFT is to refine human comments. During annotation, annotators were able to compare with the original expert comments. Additionally, we incorporated further annotations on the refined teacher comments, as shown in General Response 1, which we hope addresses your concern.

---

> ### Author Response · Authors · 2024-11-25
> **Responses to Reviewer oeU8 (2/4)**
>
> - We would kindly remind the reviewer that our criteria posted in Appendix Section B matches the widely acknowledged standards for argumentative writing, which mainly focuses on topics, argumentatives, structures, literature and expressions. For example, [3] also proposed topics, argumentations and responses in criteria for argumentation writing in education for 800 words English essay. Moreover, we manually classify the questions into the five categories and the results are shown in the following table, which are not basically conflicting with that one acknowledged in education. The authors hope to address your concerns.
>
> |      | Topics | Argumentatives | Literature | Structure | Expression |
> | ---- | ------ | -------------- | ---------- | --------- | ---------- |
> | Nums | 4      | 6              | 2          | 3         | 3          |
>
> [1] Xiaoyi Tang et al. Harnessing LLMs for multi-dimensional writing assessment: Reliability and alignment with human judgments. Heliyon. 2024.
>
> [2] Ryan Liu, Nihar B. Shah. ReviewerGPT? An Exploratory Study on Using Large Language Models for Paper Reviewing. Arxiv. 2023.
>
> [3] Omid Noroozi et al. Design, implementation, and evaluation of an online supported peer feedback module to enhance students’ argumentative essay quality. Education and Information Technologies. 2023.
>
>
>
> > W2: regarding the results are not very impressive and analysis of experiments
>
> Thank you for your suggestions regarding the methodology. We have added experiments based on Qwen2-7b-instruct on EssayC. As described in the General Response 1, RedHat outperforms the Baseline by 3.8%, 6.585%, 6.83%, and 0.136 (Hallu, Ambiguity, Info, Overall), and surpasses SFT by 12.4%, 20.00%, 34.94%, and 0.652. Additionally, the QA accuracy of Qwen reaches 92.3%. We will continue to compare with LLaMA-3.1 in English experiments, which will be included in the final version of our paper.
>
> > W2-1: small improvement
>
> The "Overall scores" represent the critique credibility, specifically their hallucinations. To clarify the effectiveness of our work, we have updated the polished human critique scores with the same annotation standard. The human overall score outperforms the best result (GLM+RedHat) by 0.06, it outperforms from the baseline(GLM) by 0.137, while directly SFT with revised instructor critiques undermines the overall score(-0.57). Thus, without introducing additional hallucinations, RedHat is closer to the level of human comments.
>
> > W2-2: not convinced by section 5.3
>
> Sorry for the confusion. We would like to clarify that there is no inherent connection between the generated critiques and the criteria-derived questions. We have adopted Reviewer rgUd's suggestion and updated the BLEURT metric as the similarity algorithm. The results are shown in the Table below. The similarity between the critique with the questions (column 1) and the answers (column 2) are presented. Comparing the baseline (Row 1) with RedHat (Rows 2-3), there is no significant change in the similarity of the question, but the critique generated by RedHat is clearly closer to the answer. This confirms that the gain in evaluation is driven by the answer inferred by the LLMs themselves, rather than by the criteria-derived question. We have corrected this confusion in our revised version.
>
> | BLEURT           | questions | Delta | Answer | Delta |
> | ---------------- | --------- | ----- | ------ | ----- |
> | Qwen             | 27.45     | 0     | 21.39  | 0     |
> | Qwen-redhat      | 25.24     | -2.21 | 31.00  | +9.61 |
> | Qwen-redhat-weak | 26.24     | -1.21 | 29.41  | +8.02 |
>
> > W2-3: the scale of the dataset is small and findings are not statistically significant.
>
> We would like to remind the reviewers that our task is to generate paragraph-level critiques for essays, and we collected 395 paragraphs from 36 essays in EssayC to be inferred by LLMs. Actually, among works related to automatic essay reviewing, works from [2] have not surpassed the number of 200 samples. The comparison with previous work can be seen in the following table. When turning to the LLM alignment dataset, datasets with fewer samples have also received wide concerns like MT-Bench(80 samples, with 2 turns, 160 in total).
>
> |                    | Studied writing amount                                       |
> | ------------------ | -------------------- |
> | RedHat: (Our Work) | 14 trained graduate TAs, 36 essays, 395 paragraphs, with critiquing on paragraph level |
> | Tang et al, 2024   | 1730 essays but only scores without critiques                |
> | Liu & Shah 2023    | 13 short papers with LLM comparison                          |
>
> As for statistical significance, we did the T-test on GLM-4 and Qwen-2 experiment overall scoring on baseline-LLM and RedHat, and got p equals to 0.048 and 0.043, indicating statistical significance to the changes.
>
> [2] Ryan Liu, Nihar B. Shah. ReviewerGPT? An Exploratory Study on Using Large Language Models for Paper Reviewing. Arxiv. 2023.

---

> ### Author Response · Authors · 2024-11-25
> **Responses to Reviewer oeU8 (3/4)**
>
> > W3: regarding the impact: task of essay feedback generation, studied language and generalising the findings
>
> Thanks for your suggestion.
>
> - On the one hand, we additionally applied our method to Qwen-2-7b-Instruct on EssayC and ChatGPT-4o and the task of English Artificial Intelligence Top Conference Paper review. Please refer to General Response 1 for result details and General Response 2 for data collection information. Results showed an apparent increase in credibility in RedHat-assisted critiques compared to the Qwen-2 and ChatGPT-4o baselines. This verified the generalizability of RedHat on English domain writing genres and different scaled LLMs.
> - On the other hand, in words of the contribution of our work:
>   - Firstly, our work has clearly defined the task of fine-grained essay critique generation, and also curated a testset with manually annotated quality results. This can serve as a benchmark for this field, as well as resources for automated evaluation development.
>   - Secondly, in terms of methodology, by leveraging the LLMs' strong QA capabilities, our approach avoids reliance on complex reasoning or simple few-shot prompt engineering. Our method outperforms SFT and post-pretraining approaches, and reveals that task-specific SFT actually introduces more hallucinations. These also validate the hallucination problem is distinct from alignment or under-fitting ones.
>   - Lastly, in terms of effects compared to previous works, we have pioneered fine-grained critiques for essays. Such critique generation methods for long essays or papers and critique quality evaluation are not discussed in previous work. Our approach is also effective across open-source 7B and 9B models as well as larger API models. In evaluating generation quality, we provide both pointwise scoring and pairwise comparison, which previous works have not yet focused on.
>
> > Minor concerns 1: the wording of hallucination
>
> In our work, LLM-generated essay critiques mainly face two types of problems. Type I: Hallucinate Fantasizing writing problems that do not exist in the essay. Type II: Overlook the essay's logic and structure by overemphasizing on details, or just vague comments with no in-depth advice. We find the two types are consistent with hallucinations discussed in previous literatures. Specifically speaking,
>
> - For type I, [1] mentioned "ignore the source material altogether" as *Extrinsic Hallucination* in summarization task. [2] defined "generation of nonfactual, untruthful information, especially ignoring the context"  in ChatGPT's reasoning and multimodal tasks.
> - For type II, [1] mentioned "misrepresent information from the doc" as *Instrinsic Hallucination* in summarization task, [3] concluded "Generation of plausible looking statements that are factually incorrect" in conversation task.
>
> Considering the features of our type I and II hallucinations, our definition agrees with hallucination definition in summarization, conversation, QA, reasoning tasks. Based on the discussion above, we lend the wording **hallucination** to describe our phenomenon. We present the discussed literature in the following Table 5 for further interest.
>
> | Work                 | Citation | Task                              | Definition                                                   |
> | -------------------- | -------- | --------------------------------- | ------------------------------------------------------------ |
> | Maynez et al., 2020  | 1180     | Summarization (RNN)               | (On Page 3) Intrinsic Hallu: misrepresent info of from the docExtrinsic Hallu: Model ignore the source material altogetherFactual Hallu: Contain info not found in doc ( composed of instrinsic and extrinsic hallu) |
> | Shuster et al., 2021 | 592      | Knowledge Conversation (BART, T5) | (On Page 1) Generation of plausible looking statements that are factually incorrect |
> | Bang et al., 2023    | 1301     | QA, reasoning (ChatGPT)           | (On Page 3) Generation of nonfactual, untruthful information, mainly extrinsic in Work 1 |
>
> [1] Joshua Maynez et al. On faithfulness and factuality in abstractive summarization. ACL. 2020
>
> [2] Kurt Shuster et al. Retrieval augmentation reduces hallucination in conversation. EMNLP. 2021
>
> [3] Yejin Bang et al. A multitask, multilingual, multimodal evaluation of chatgpt on reasoning, hallucination, and interactivity. IJCNLP. 2023

---

> ### Author Response · Authors · 2024-11-25
> **Responses to Reviewer oeU8 (4/4)**
>
> > Minor concerns 2: strong vocabulary for contributions
>
> We apologize for the inaccurate statement. In our revised version, we have corrected it to "suggest" and added the Qwen-2-7b experiment on EssayC as well as English experiments on ChatGPT-4o. Our results demonstrate that our approach is effective in: (a) different base models (GLM4, Qwen2, ChatGPT-4o), (b) the same base model with and without SFT, and (c) different types of long writings and languages such as undergraduate essays and conference papers.
>
> Also, we kindly remind the reviewer that in Table 4, the horizontal axis represents the evaluators (GPT4 for auto method, human for manual method) of the quality of the polished paragraphs. The number ~60% reveals that GPT evaluates them as either equally good or equally bad. We conclude that both humans and GPT prefer the polished paragraphs based on RedHat critique.
>
> > Recommendations (a): root more firmly in existing literatures
>
> We are very grateful to follow your kind and detailed suggestions, which essentially improved the quality of our work. We concluded the corresponding improvements to them.
>
> We summarize the aforementioned literature discussion and included it in the revised version. [1] directly provides a score without critiques, hard to improve a 5000 characters essay; [2] offers critiques that do not address the improvement details, lacking paragraph-level feedback and compromising evaluation; [3] proposes topics, argumentations and responses in criteria for argumentation writing in education, and our evaluation criteria and derived questions are not conflicting with it. The above work's list of evaluation criteria differs from our work for the differences between essays and conference paper purposes. We have also added a few-shot method as a practicable baseline to our experiments. Thank you for your suggestion; we have added the few-shot method to the main experiment table.
>
> [1] Xiaoyi Tang et al. Harnessing LLMs for multi-dimensional writing assessment: Reliability and alignment with human judgments. Heliyon. 2024.
>
> [2] Ryan Liu, Nihar B. Shah. ReviewerGPT? An Exploratory Study on Using Large Language Models for Paper Reviewing. Arxiv. 2023.
>
> [3] Omid Noroozi et al. Design, implementation, and evaluation of an online supported peer feedback module to enhance students’ argumentative essay quality. Education and Information Technologies. 2023.
>
> > Recommendations part (b): increase the scope
>
> We have added the results of experiments of Qwen2-7b on EssayC and ChatGPT-4o on English Conference papers. RedHat outperforms the baseline, SFT, few-shot, etc., in overall performance, hallucination, ambiguity, informativeness, and pairwise comparison. This demonstrates that RedHat reduces hallucinations and improves the credibility of the critiques. We have also updated the experiment section to include these results in our revised manuscript.
>
> To automatically score overall/hallucination/ambiguity/informativeness, we have trained 4 Linear Regression Models by using the GLM4 and Qwen2 experimental data as a score predictor for response quality. We made a random split between the training set and the validation set with a ratio of 9:1. The regression fitting accuracies are shown in the table below. Results showed that the fitting in the hallucination dimension is not good, indicating our view that SFT alone does not contribute to hallucination relief, and echoing the issues in our paper. Thanks for your suggestion and we will release this score model and consider using RLHF in future work to further improve scoring quality.
>
> |           | Overall (6 class) | Hallu(2 class) | Ambiguity(2 class) | Informativenss(3 class) |
> | --------- | ----------------- | -------------- | ------------------ | ----------------------- |
> | GLM-4-9b  | 0.73              | 0.64           | 0.74               | 0.71                    |
> | Qwen-2-7B | 0.76              | 0.63           | 0.79               | 0.74                    |

---

> ### Author Response · Authors · 2024-11-29
> **We look forward to your feedback**
>
> Dear Reviewer oeU8,
>
> The authors are grateful for your thought-provoking suggestions. Since there are three days left until the discussion period ends, we are unsure whether the responses above has fully addressed your concerns. If there are any remaining issues, please feel free to contact us. We eagerly look forward to your further comments.
>
> Best regards,
>
> The authors

---

> > ### Comment · Reviewer_oeU8 · 2024-12-03
> > **still not convinced**
> >
> > I thank the authors for the overly detailed and wordy response. My main criticisms still persist:
> >
> > The work is not firmly rooted in literature. We computer scientists didn't invent argumentation. I suggested to compare the induced questions with argumentation literature, the authors provide three references involving LLMs or computational methods, which in my view miss the point.
> >
> > I appreciate the inclusion of other models which largely seem to confirm my point that the impact of RedHat is limited. It is almost too convenient that compared with other models (for which by the way I do not reach the same conclusion that "human prefer redhat" - the correct way would be to say "for most cases, humans have no preference, but if they do they slightly prefer RedHat over baseline generations") RedHat with GPT4 so significantly improves the scores. A reasonable follow-up question would be - why so? Why does it work so well for GPT4 and not for the other models?
> >
> > My assessment regarding the gravity of the contribution still persists.

---

> ### Author Response · Authors · 2024-12-03
> **Further discussion on root literature and contribution (1/2)**
>
> We appreciate your in-depth concerns and criticisms of rooting literature and the human preference part results. They help us reflect on the delivery of our work and improvements.
>
> > Connection between induced questions with existing literatures.
>
> To concisely address your concerns, we have reached the Delphi Report [1] (for the purpose of educational assessment), PISA Report [2] (for critical thinking framework), and [3] (for high-school essay assessment in education) in the field of education. We briefly list the criteria from each work. **We have sorted the suggested dimensions from the two reports and manually classified the induced questions in Appendix E.1 into the slots, as the following table shows.** The induced questions have covered almost all dimensions in acknowledged argumentative evaluation criteria from the perspective of pedagogy and creative-thinking. We hope that such analysis addresses your concerns on rooting literature on argumentation essay evaluation criteria.
>
> | Delphi Report                  | **Content** | **Structure and Organization** | **Evidence and Support** | **Argumentation** | **Style and Voice** | **Critical Thinking Dispositions** | **Domain-Specific Knowledge** | Leftover questions|
> | ------------------------------ | ----------- | ------------------------------ | ------------------------ | ----------------- | ------------------- | ---------------------------------- | ----------------------------- | --------- |
> | Overlap with Induced Questions | 3           | 3                              | 3                        | 2                 | 3                   | 2                                  | 2                             | 0         |
>
> | PISA                           | **Idea Generation/innovation** | **Idea Evaluation and Improvement** | **Contextual Factors** | **Domain-Specific Considerations** | Leftover questions|
> | ------------------------------ | ------------------------------ | ----------------------------------- | ---------------------- | ---------------------------------- | --------- |
> | Overlap with Induced Questions | 4                              | 3                                   | 6                      | 2                                  | 3         |
>
> | Omid Noroozi et al, 2023       | **Introduction on the topic** | **Taking a position on the topic** | **Arguments for the position** | **Justifications for arguments for the position** | **Arguments against the position (counter-arguments)** | **Justifications for arguments against the position** | **Response to counter-arguments** | **Final conclusion and implications** | Leftover questions|
> | ------------------------------ | ----------------------------- | ---------------------------------- | ------------------------------ | ------------------------------------------------- | ------------------------------------------------------ | ----------------------------------------------------- | --------------------------------- | ------------------------------------- | --------- |
> | Overlap with Induced Questions | 2                             | 2                                  | 6                              | 4                                                 | 1                                                      | 0                                                     | 1                                 | 1                                     | 1         |
>
> One more word, in our manuscript, we have listed our criterion in Appendix C for review and it mainly contains **topic, literature, arguments, structure, language and norms**. We have also made the classification of induced questions in responses (2/4). In fact, the above examinations are ensured by graduate teaching assistants during experiments.
>
>
> [1] Facione. (1990). Critical thinking: A statement of expert consensus for purposes of educational assessment and instruction (The Delphi Report).
>
> [2] Organisation for Economic Co-operation and Development. (2023). PISA 2022 Assessment and Analytical Framework. OECD Publishing.
>
> [3] Noroozi et al. (2023). Design, implementation, and evaluation of an online supported peer feedback module to enhance students’ argumentative essay quality. *Education and Information Technologies*, *28*(10), 12757-12784.

---

> ### Author Response · Authors · 2024-12-03
> **Further discussion on root literature and contribution (2/2)**
>
> > Result convincingness.
>
> As for the follow-up questions on the machenism of methodology, we attribute the success of RedHat due to the better contextual digest ability of ChatGPT-4o than <10 B models and better DocQA accuracy. ChatGPT-4o is better at digest the highlights from the answers, thus contributing to the critique. Also, we adopted the suggestions from Reviewer rhUd and posted the detailed **inter-annotator agreement for each score** in the main experiment Table. Results show moderate to substantial significance rather than randomness. **Please refer to [Further discussion on Inter-Annotator Agreement](https://openreview.net/forum?id=IULlNTZZel&noteId=vb6vPAKCma) under Reviewer rhUd' comment for details**. We further provide the **significant-test (T-test)** of the results between the base-LLM model and LLM+RedHat results in the following Table. The possibilities for rejection of significance are all lower than 0.05.
>
> | Pre- and post- data | GLM & RH | Qwen & RH | GPT & RH  |
> | ------------------- | -------- | --------- | --------- |
> | P value             | 0.048    | 0.043     | 5.887e-10 |
>
> > Contribution of the paper.
>
> Basically, we kindly remind the reviewer that our core contribution lies in the **reduction of hallucination** in generated critiques, and results from Table 4 - main experiment - showed reduction in hallucination dimension. While it is a controversial topic whether human preference equals factuality correctness as many recent works in LLM alignment shows, we provide the comparison mainly to indicate that the **hallucination reduction might contribute to better preference by human**. We respect your suggestion for meticulousness in claiming preference and are willing to make further revisions to the manuscript.
>
> We briefly echo the structure of our validation of RedHat:
>
> 1.  We have analyzed the similarity between the induced questions and answers in the above response (2/4) and Table 5 in Section 5.3 from the manuscript, showing the answer-similarities are correlated to the reduction in hallucination.
> 2. Further, the answer accuracy rate for GLM, Qwen, GPT is 85.5%, 92.2%, 95.6% respectively (in Section 5.3 from the manuscript), correlates to the dynamics in the results in Table 4 scores in the major dimensions.
> 3. We suggest that RedHat helps reduce hallucination in the above mechanism, since RedHat has relieved the core source of hallucination that base-LLM alone is not faithfully grounded on the target essay.
>
> We thank you again for your efforts for reading the lengthy responses. We hope the above materials can solve your concerns and assist you with reevaluating the contributions of RedHat, especially in the aspects of the resources and discussion on the credibility of automatic evaluation for human-written essays - the core motivation of our work.

---

### Author Response · Authors · 2024-11-25
**General Response 1: Supplemental Experiment of RedHat on other LLMs and other languages/datasets (1/2)**

> Regarding whether RedHat can be generalized to other LLMs, languages, datasets

We are grateful for all reviewers' suggestions on consolidating the experiment scope.

First, we have followed reviewer ssM2's suggestion, changing the naming of one of our evaluation dimensions "**Detailedness**" into "**Ambiguity**", since it is lower-is-better in the results. We also revised the sign in the objective function.

In order to address Reviewer oeU8 and ssM2' concern about the findings can be applied to other languages or LLMs, **we conduct two** **additional** **experiments**: **Qwen2-7b-Instruct on EssayC** and **ChatGPT-4o on English** publicly available artificial intelligence conference papers (e.g. ICLR, Neurips, ACL). Conclusions are consistent with GLM-4-9b on EssayC that RedHat reduces hallucinations and improves the overall credibility in generated critiques. Details are as follows.

**Qwen2-7b-Instruct on EssayC**

For the experiment on Qwen2, RedHat outperforms the Baseline Qwen2 by 0.136/3.8%/6.585%/6.83% (Overall, Hallu, Ambiguity, Info), and surpasses SFT by 0.652/12.4%/20.00%/34.94%. We also evaluated polished human instructor's critiques under the same standards and revealed 0.200/15.19%/3.03%/18.73% improvement to baseline Qwen2. Additionally, the QA accuracy of Qwen reaches 92.3%, which indicates that LLM is more understood in the evaluated essay, thus contributing to the reduction in hallucination, ambiguity and increments in informative and overall scores.

|                   | Overall ($\uparrow$) | Hallucination % ($\downarrow$) | Ambiguity % ($\downarrow$) | Informative % ($\uparrow$) |
| ----------------- | -------------------- | ------------------------------ | --------------------------- | --------------------------- |
| Human             | 3.387                | 47.34                          | 11.65                       | 30.63                       |
| Qwen2             | 3.187                | 62.53                          | 14.68                       | 11.90                       |
| 5-shots           | 3.178                | 61.01                          | 11.14                       | 12.66                       |
| Qwen2-redhat      | 3.267                | 62.03                          | 7.59                        | 15.70                       |
| Qwen2-redhat-weak | 3.323                | 58.73                          | 8.10                        | 18.73                       |
| Qwen2-sft         | 2.615                | 74.43                          | 27.59                       | -19.24                      |
| Qwen2-sft-redhat  | 2.636                | 77.72                          | 22.28                       | -10.63                      |
| Qwen2-PT          | 2.777                | 71.65                          | 24.30                       | -9.11                       |

In pairwise comparison, RedHat wins over Baseline Qwen2 by 7.22% and 10.36% with GPT-generated answers and self-generated answers.

|                   | RedHat better | Qwen better | Both are good | Both are bad |
| ----------------- | ------------- | ----------- | ------------- | ------------ |
| Qwen2-redhat      | 33.26         | 26.04       | 7.38          | 33.32        |
| Qwen2-redhat-weak | 36.30         | 25.94       | 7.18          | 30.57        |

---

> ### Author Response · Authors · 2024-11-25
> **General Response 1: Supplemental Experiment of RedHat on other LLMs and other languages/datasets (2/2)**
>
> **[2/2] ChatGPT-4o on Artificial Intelligence Conference Paper**
>
> To consolidate the validity of our work, we followed [1] and picked a subset of ICLR, Neurips, ACL Conference papers and arxiv papers amounting to 10 with 100 paragraphs to be critiques. We intentionally chose picked those papers whose authors' were accessible so that they could judge the quality over the generated critiques.
>
> **Implementation**: We applied the ICLR reviewer guidelines as evaluation criteria. Since ICLR reviewer guidelines have already contained more than 10 questions in it, we replace the guideline questions with description of the expectation for a good conference paper on those questions.
>
> **Evaluation**: to ensure the effectiveness of human annotation, we contacted and invited the authors of the papers to grade along with Hallucination, Ambiguity, Informative, Overall scores along with comparison. The results are shown in the following table. Each critique has received two authors' judgments and the inter annotation agreement reaches 0.74.
>
> **Results**: results showed an apparent increase in credibility in RedHat assisted critiques. This verified the generalizability of RedHat on English domain writing genres and LLMs.
>
> |                   | Overall ($\uparrow$) | Hallucination % ($\downarrow$) | Ambiguity % ($\downarrow$) | Informative % ($\uparrow$) |
> | :---------------: | :------------------: | :----------------------------: | :-------------------------: | :-------------------------: |
> | ChatGPT-4o        | 2.448                | 76.92                          | 11.99                       | -9.99                       |
> | ChatGPT-4o-RedHat | 3.549                | 42.96                          | 8.99                        | 23.98                       |
>
> |          | ChatGPT-4o Better | ChatGPT-4o-RedHat Better | Both good | Both bad |
> | -------- | :---------------: | :----------------------: | :-------: | :------: |
> | Win rate | 7                 | 58                       | 10        | 25       |
>
> We will continue to compare with LLaMA-3.1 in English experiments to further validate the effectiveness in English of RedHat, which will be included in the final version of our paper.
>
> [1] Ryan Liu, Nihar B. Shah. ReviewerGPT? An Exploratory Study on Using Large Language Models for Paper Reviewing. Arxiv. 2023.

---

### Author Response · Authors · 2024-11-25
**General Response 2: Information and description of EssayC**

> Regarding Information about EssayC annotation quality and statistics

Reviewer ssM2 and rgUd have expressed their interests in the detailed information about EssayC. Here we sincerely introduce the construction process and statistical overview.

- **Essay Selection**: EssayC randomly collects undergraduate essays whose topics cover Environment Science, Biological Science, Software Engineering, Game Industry, Earth, Social Science, Journalism and Communications, Economics, Humanities, Literature comments, and so on. Most science, engineering and humanity and social science are covered.
- **Polish**: Human comments may be incomplete in grammar and organization. We used GLM4-130B to refine and complete their grammar and structure based on the human comments.
- **Testset split**: 36 essays are randomly picked out of the above process under each field topics. The leftovers are beneficial as training data for supervised fine-tuning.
- **Filter**: For the testset, we asked the annotators to read through the teacher's critiques in the paragraph and filter out unqualified ones(like with only punctuation marks or subjective comments with mere expressions of feeling). Then we devised a raw critique-quality classifier on GLM-4-9B to auto-filter the leftovers in the train data section. Critique numbers drop from 675 to 395 in the test set, and from 51238 to 31694 in the training set after filtration.
- **Annotation and quality assessment**: During the evaluation phase, each generated comment is annotated by two graduate teaching assistants. In case of discrepancies, a third graduate teaching assistant makes the final decision. Our overall Inter Annotator Agreement is 0.71 in GLM-4 and Qwen-2 as a whole, ensuring annotation consistency and reducing random interference.
- **English dataset**: To consolidate the gravity of our work, we followed [1] and picked a subset of ICLR Conference paper and arxiv papers with the number 10. We picked those papers containing less formulas and illustrations, and more importantly, ensuring the paper authors' are accessible so that they could judge the quality over the generated critiques.

We have updated the information about EssayC in Section 2.

|                                 | EssayC | English |
| ------------------------------- | ------ | ------- |
| Essay                           | 36     | 10      |
| Avg len                         | 5204.7 | 42087.3 |
| Critiqued paragraphs            | 395    | 100     |
| Avg paragraph  length           | 278.2  | 1278.4  |
| Avg teacher critique length     | 76.78  | /       |
| Pointwise Annotations           | 5530   | 200     |
| Pointwise Annotation Dimensions | 4      | 4       |
| Pairwise Annotations            | 1580   | 100     |
| Avg Critique len                | 98.53  | 89.65   |

[1] Ryan Liu, Nihar B. Shah. ReviewerGPT? An Exploratory Study on Using Large Language Models for Paper Reviewing. Arxiv. 2023.

---

### Author Response · Authors · 2024-11-26
**General Response 3: Manuscript Update**

We sincerely thank all reviewers for their constructive suggestions, which have greatly helped improve our manuscript. In response to each reviewer’s feedback, we have made revisions to enhance clarity, precision, and depth. Below is a summary of the major changes made in the revised version:
- Section 1 - Introduction: We have revised the wording in Line 101, changing "prove" to "suggest" to better reflect the tone of our argument.
- Section 2 - Task & Dataset Definition: We have added Table 1, which links our work to existing literature. Additionally, we revised Lines 126-140 to align with these changes, and added Lines 142-156 to introduce the construction and statistics of the EssayC dataset, with the addition of Table 2.
- Section 3 - Hallucinations: We modified Lines 187-196 to better connect our definition of hallucination to related work in other tasks. In Lines 177 and 179, we replaced "Detailedness" with "Ambiguity" and updated the notation "A(c)" and its corresponding sign in the objective function.
- Section 4 - Methodology: We clarified the statement of RedHat’s contribution in Lines 245-253. Additionally, in Line 291, we replaced "Detailedness" with "Ambiguity" to ensure consistency.
- Section 5 - Experiments:
  - In Section 5.1 (Lines 319-323), we provided a more detailed description of the datasets used in our experiments.
  - In Line 358, we included a description of the few-shot method employed.
  - In Line 365, we again replaced "Detailedness" with "Ambiguity."
  - In Lines 374-376, we added the annotation for Inter-Annotator Agreement and included a discussion of quality control in the annotation process.
  - We updated Table 4 and Figure 4 to include results for Qwen2 and ChatGPT4.
  - In Section 5.3, we introduced additional evaluation metrics, including BLEURT and BERTScore, and included Qwen2 results for both questions and answers.
- Section 6 - Conclusion: The discussion of limitations has been moved to Appendix B for clarity and space.
- Appendix:
  - We added the criteria for critiquing English conference papers to Appendix C.2.
  - We included the questions for English conference paper critiquing in Appendix E.2.
  - We provided detailed descriptions of the data preparation for SFT and PT in Appendices F.1.1 and F.2.1.
  - Finally, we added the few-shot prompt to Appendix F.3.

We believe these revisions address the reviewers’ concerns and significantly improve the manuscript. Thank you once again for your valuable feedback.

---

### Meta-Review · Area_Chair_NKsD · 2024-12-20

**Metareview:**

The paper proposing a prompting technique to generate grounded critiques for essays; the main goal is to mitigate "hallucinations" in the critiques that refer to phenomenon not seen in the essay. The approach shows some improvement over baselines like SFT, etc. based on human evaluation. Reviewers raise the issue of this human evaluation criterion not being grounded on existing literature in argumentation. The paper can be augmented with more discussion/analysis of the critiques to better outline how meaningfully different the generated critiques are (score differences of <0.5 points on a likert scale are not informative in this regard).

**Additional Comments On Reviewer Discussion:**

Reviewer oeU8 raises questions about the criterion used to evaluate the generated critiques, and whether these are grounded in argumentation literature. Although the authors present a response, the reviewers (including the area chair) are not convinced that the evaluation is sound. Furthermore, it is known that human annotators, unless specifically trained, often conflate different quality dimensions. To allay these criticisms, a more detailed analysis of how the critiques change qualitatively is needed.

 Reviewer ssM2 asked some clarification questions about the training setup and details about the dataset, that was addressed in the response. Their main criticism was that the contribution of the work is incremental.

---

### Decision · Program_Chairs · 2025-01-22

Reject